# YOUR THOUGHTS TELL WHO YOU ARE: CHARACTERIZE THE REASONING PATTERNS OF LRMS

## ABSTRACT

Current comparisons of large reasoning models (LRMs) focus on macro-level statistics such as task accuracy or reasoning length. Whether different LRMs *reason* differently remains an open question. To address this gap, we introduce the LLM-proposed Open Taxonomy (LOT), a classification method that uses a generative language model to compare reasoning traces from two LRMs and articulate their distinctive features in words. LOT then models how these features predict the source LRM of a reasoning trace based on their empirical distributions across LRM outputs. Iterating this process over a dataset of reasoning traces yields a human-readable taxonomy that characterizes how models think. We apply LOT to compare the reasoning of 12 open-source LRMs on tasks in math, science, and coding. LOT identifies systematic differences in their thoughts, achieving 80–100% accuracy in distinguishing reasoning traces from LRMs that differ in scale, base model family, or objective domain. Beyond classification, LOT's natural-language taxonomy provides qualitative explanations of how LRMs think differently. Finally, in a case study, we link the reasoning differences to performance: aligning the reasoning style of smaller Qwen3 models with that of the largest Qwen3 during test time improves their accuracy on GPQA by 3.3–5.7%.

## 1 INTRODUCTION

Following the success of GPT-o1 and DeepSeek-R1, a wave of large reasoning models (LRMs) has recently become available. These models differ in training recipes and report varying benchmark performance, but far less is known about whether they also reason differently from one another. In this paper, we pose a fundamental question: can LRMs be distinguished by their reasoning patterns, and if so, what are the key distinguishing traits?

A growing body of work has begun probing the reasoning style of individual LRMs, yielding insights into how artificial thinkers "think". Marjanović et al. (2025), for example, finds that DeepSeek-R1's reasoning depth correlates with the human cognitive load when processing complex sentences. Bogdan et al. (2025) annotates functions of DeepSeek-R1-Distill-Qwen's reasoning steps, showing that plan generation and re-evaluation are critical for solving math problems.

However, only a few studies attempt broader comparisons across LRMs and on multiple reasoning behaviors. Gandhi et al. (2025) compares the reasoning patterns of base large language models (LLMs) and their RL fine-tuned variants, finding that the habits of base models correlate with fine-tuning gains. Along the same lines, Jiang et al. (2025) shows that LRMs differ in how they structure their reasoning steps. However, Gandhi et al. (2025); Jiang et al. (2025); Bogdan et al. (2025) adopt a *deductive* approach, relying on fixed, researcher-defined taxonomies of reasoning behaviors when comparing artificial thinkers. The deductive approach risks biasing analyses towards researchers' theories and overlooking unexpected behaviors of models, such as attempting to "visualize" the chemical structure of compounds given in the question.

To address this limitation, we introduce the LLM-proposed Open Taxonomy (LOT), an *inductive* method that identifies reasoning features distinguishing two LRMs directly from their outputs. LOT operates in three stages: (1) an LLM compares thinkings from two LRMs on the same question and highlights distinguishing reasoning traits in natural language; (2) the LLM annotates these features in reasoning traces from other questions, converting textual reasoning into vectors of features; (3) a logistic classifier is trained on these vectors to predict the source model for unseen traces. When

classification fails on a new trace, LOT returns to stage 1 to propose new features observed in the failed sample. Iterating this cycle yields an open taxonomy of reasoning traits that reliably separates the thought processes of different LRMs.

We apply LOT to compare and classify the reasoning traces of 12 LRMs across diverse model scales, base model families, and specialized domains. LOT achieves 80-100% accuracy in classifying reasonings of LRMs when they differ substantially along one of the axes above. In classifying LRMs with various parameter scales, LOT outperforms few-shot prompting (by 23.8% on average), a recent automatic prompt engineer method, VML (Xiao et al., 2025) (by 19.6%), and a human-defined reasoning taxonomy (Gandhi et al., 2025) (by 11.7%) in accuracy.

Its natural-language taxonomies also provide verbal explanations of systematic differences between LRMs, such as a smaller model's tendency toward circular reasoning or a code-specialized model's usage of Python functions to solve math problems. As a case study, we further link the reasoning differences among Qwen3 models at different scales with their performance gaps on GPQA-Diamond, showing that modifying the smaller Qwen3 models' reasoning styles can improve their accuracy by 3.3–5.7%.

In summary, our main contributions are: (1) introduce LOT, an inductive method that constructs human-readable taxonomies of reasoning features to characterize the thought processes of LRMs; (2) show that LOT accurately classifies LRMs' reasoning across domains and outperforms existing approaches; (3) explain systematic reasoning differences in natural language; and (4) through a case study on Qwen3 models, demonstrate that these differences have a causal link to performance gaps.

## 2 RELATED WORK

**Classification as a Probe to Illustrate the Models' Behavioral Differences**   Existing studies of LRMs' reasoning (Gandhi et al., 2025; Jiang et al., 2025; Bogdan et al., 2025) rely on predefined taxonomies of behaviors, limiting their analyses to researcher-chosen categories. Recently, Sun et al. (2025) uses classification as an exploratory probe to detect differentiating output patterns of non-reasoning LLMs, such as Grok-2 and Gemini-1.5. They train neural classifiers to predict the source model of generated texts and obtain high accuracies that suggest the existence of "signatures" patterns in LLMs' outputs.

However, the features learned by neural models are not directly interpretable. Sun et al. (2025) instead infers the LLMs' behavioral differences through post-hoc counterfactual intervention, which manipulates specific textual properties chosen by the researchers and measures the change in classification accuracy. Since the intervened properties are chosen by researchers, this feature-discovery process remains deductive and may not reflect what the classifier has learned.

Can we use the classifier's learned features to directly explain the LLMs' behavioral differences? In this work, we use classification not only as an *exploratory* sensor to detect reasoning differences, but also as an *explanatory* tool to interpret what these differences are. To achieve this, we design a novel automatic prompt engineer algorithm that, by comparing the thought processes of LRMs, inductively generates human-readable reasoning features for classifying those processes.

**Automatic Prompt Engineer for Interpretable Text Classification**   Recent Verbalized Machine Learning (VML) (Xiao et al., 2025) proposes using LLMs to generate interpretable, natural-language decision trees for text classification. In VML, the LLM receives a batch of training samples as input and updates the decision rules, expressed in natural language, based on the observed patterns. VML generates a decision tree by iterating this process. While effective for short-text tasks such as classifying word–gender associations (Srivastava et al., 2023), VML is impractical for classifying long reasoning traces, which may span tens of thousands of tokens. To accommodate the context window of existing LLMs, VML must drastically reduce its batch size, leaving updates to its decision tree unstable and sensitive to noise.

Other automatic prompt engineer (APE) methods (Zhou et al., 2022; Guo et al., 2024; Benara et al., 2024; Pryzant et al., 2023) can generate a classification instruction without batched examples, but they rely on an initial pool of candidate instructions. The initial instructions are crafted either by humans or from an LLM's prior knowledge of the task. However, given the recency of the LRMs we studied, neither we nor recent models have reliable knowledge about their reasoning patterns.

---

**Algorithm 1** LLM-proposed Open Taxonomy

---

**Require:** $\mathcal{D}_{\text{train}} = \{(a, b, y_a, y_b)\}_n$: paired reasonings from two LRMs on the same questions, $M_\theta$: LLM annotator
1: Annotate distinguishing features $\{c_i\} \leftarrow M_\theta(c_1, \ldots, c_m \mid y_a, a, y_b, b)$ observed in a sample
2: Initialize $\mathbb{C} \leftarrow \{c_1, \ldots, c_m\}$        subsection 3.1
3: **while** not converged **do**
4:      Sample $\mathcal{D}_{batch} \subset \mathcal{D}_{\text{train}}$.
5:      **for** $(a, b, y_a, y_b) \in \mathcal{D}_{batch}$ **do**
6:          Encode $a$, $M_\theta(a_{c_1}, \ldots, a_{c_{|\mathbb{C}|}} \mid \mathbb{C}, a)$
7:          Update encoding $\boldsymbol{a}_\mathbb{C} = \langle a_{c_1}, \ldots, a_{c_{|\mathbb{C}|}} \rangle$ to $A_\mathbb{C}$; Repeat for $b$      subsection 3.2
8:      **end for**
9:      **while** $\mathbb{C}$ unchanged **&** not converged **do**
10:         Train logistic classifier $\phi : x_\mathbb{C} \mapsto y_x$ on $\{A_\mathbb{C}, B_\mathbb{C}\}$
11:         Encode $(a, b) \sim D_{train}$
12:         Predict $\hat{y}_a, \hat{y}_b$ using $\phi$
13:         **if** $(\hat{y}_a, \hat{y}_b) \neq (y_a, y_b)_i$ **then**
14:            Annotate additional features $M_\theta(\hat{\mathbb{C}} \mid \mathbb{C}, a, b)$
15:            Update taxonomy $\mathbb{C} \leftarrow \mathbb{C} \cup \hat{\mathbb{C}}$      subsection 3.3
16:         **else**
17:            $A_\mathbb{C} \leftarrow A_\mathbb{C} \cup \{\boldsymbol{a}_\mathbb{C}\}$, $B_\mathbb{C} \leftarrow B_\mathbb{C} \cup \{\boldsymbol{b}_\mathbb{C}\}$
18:         **end if**
19:      **end while**
20: **end while**
21: **return** $\mathbb{C}, \phi$

---

## 3   METHOD: LLM-PROPOSED OPEN TAXONOMY

We hence seek a different approach to classifying reasoning traces that (1) can generate classification features directly from reasoning data without relying on predefined candidates and (2) can refine these features without requiring batched inputs that exceed LLM context limits.

To meet these criteria, we introduce LOT, an APE method that builds an open taxonomy of human-readable reasoning features for classifying reasoning traces from different LRMs. LOT is inspired by the inductive coding process in qualitative research: instead of starting from predefined categories, it derives candidate reasoning features directly from observed reasoning data. These features are expressed in natural language, applied to annotate new traces, and continuously refined so that reasoning traces from different LRMs can be reliably distinguished by their annotations.

In the following subsections, we describe how LOT proposes reasoning features from limited examples and produces a reliable classification model without requiring batched reasoning inputs.

### 3.1   INITIALIZATION OF LOT

We do not assume any prior knowledge about the reasoning differences between two LRMs. At initialization, we input the LLM $M_\theta$ with a pair of reasoning traces $(a, b)_i$ from two LRMs $A$ and $B$ that solve the same question, along with labels indicating their respective source models $(y_a, y_b)$. The taxonomy $\mathbb{C}$ is initialized with the distinguishing reasoning features $\{c_1, \ldots, c_m\} \leftarrow M_\theta(c_1, \ldots, c_m \mid y_a, a, y_b, b)$ identified from this pair.

### 3.2   ENCODING AND CLASSIFICATION WITH LOT

After obtaining an initial $\mathbb{C}$, we represent new reasoning traces within the feature space spanned by the LLM-proposed reasoning traits. Encoding is done by instructing the LLM to annotate the occurrence of each reasoning features $c$ in the trace following the $c$'s natural-language definition.

We tested two representations of reasoning traces: presence of reasoning (PoR) and bag of reasoning (BoR). PoR represents a reasoning trace as a binary vector with each dimension representing the

presence or absence of a reasoning feature $c \in \mathbb{C}$. BoR is generated by annotating the function of each sentence in the trace, taking into account the frequency of reasoning behaviors.

To classify a reasoning trace $x$, we first annotate sampled reasoning traces from models $A$ and $B$ to construct a dataset of vectors representing the two models' reasoning, $\{A_{\mathbb{C}}, B_{\mathbb{C}}\}$. We then train a logistic regression classifier $\phi$ that maps $\{A_{\mathbb{C}}, B_{\mathbb{C}}\}$ to their source models. For a new reasoning trace, we annotate it using the same $\mathbb{C}$ and predict its source LRM through $\phi$.

## 3.3 Iterative Updates of LOT

The reasoning differences observed in one pair of traces during initialization may not be sufficient for classifying other samples. We improve the separability of reasoning traces in LOT by iteratively expanding its feature dimension.

To do so, we apply the trained $\phi$ and $\mathbb{C}$ to new reasoning pairs sampled from the training set. When classification fails, it suggests that the feature set is potentially incomplete. For the failed sample, we provide the source LRM labels of the two traces and instruct the LLM to propose additional reasoning differences $\hat{\mathbb{C}}$.

After $\mathbb{C}$ is updated, LOT returns to annotate another batch of samples using the new $\mathbb{C}$. We combine the new encodings with the existing vector dataset by expanding its dimensions and imputing the missing values. For PoR encodings, we impute the missing values with 0. For BoR encodings, we find that KNN imputation (Emmanuel et al., 2021) provides more stable classification performance during training. Finally, the logistic classifier $\phi$ is re-trained on the updated vector dataset. The imputation is applied only during training. To avoid artifacts from missing or imputed values, all behavioral analyses in the following section use annotated traces from the test split.

**Iteration and Convergence** Training iterates the feature generation, encoding, and update steps described above. It is converged when no changes are made to the taxonomy for $N = 20$ consecutive iterations or when it reaches the maximum of $M = 2|\mathcal{D}_{\text{train}}|$ training samples.

## 3.4 Comparison with Existing Approaches

Our method differs from existing APEs and deductive analyses in its outcome, generation of classification program, and open feature set.

**Outcome** Original APE (Zhou et al., 2022) and its variants such as ProTeGi (Pryzant et al., 2023) and EvoPrompt (Guo et al., 2024) aim to identify the best-performing prompt by narrowing down a set of candidate classification instructions. In contrast, LOT improves the classifiability of reasoning traces by expanding a set of reasoning features.

**Classification Program Generation** LOT separates the generation of classification features and their parameters across different forward passes. In one forward pass, LOT proposes predictive reasoning features. In subsequent passes, it calibrates the parameters of these features based on annotated traces. VML (Xiao et al., 2025) requires the LLM to analyze a batch of data and produce classification rules within a single forward pass, which is impractical for long texts.

**Open Feature Set** LOT keeps its set of reasoning features open during training, iteratively *expanding* it as the LLM observes more reasoning data. Deductive studies rely on a fixed taxonomy defined by researchers before analysis. APE methods also require an initial pool of candidate instructions.

# 4 Characterizing the reasoning patterns of LRMs

We apply LOT to classify reasoning traces from 12 open-source LRMs that vary in parameter scales, base model families, and task specializations. Our goal is to understand whether these model differences would lead to systematic differences in LRMs' reasoning, and if so, what are they?

All classifications are performed *pairwise* (binary) between two LRMs with their reasoning traces on the same dataset. We cover five datasets: GPQA-Diamond (Rein et al., 2024) for graduate-level

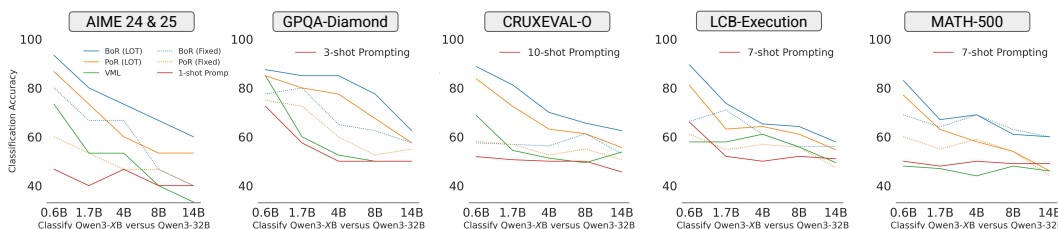

Figure 1: Test accuracies in classifying the reasoning traces generated by Qwen3-32B and one of its smaller variants. Dotted lines indicate accuracies based on BoR and PoR encodings generated using a **fixed**, human-defined reasoning taxonomy (Gandhi et al., 2025) (see Appendix G for details).

science reasoning; MATH-500 (Hendrycks et al., 2021) and AIME-24/25 AIME (2025) for high school competition math; and CRUXEVAL (Gu et al., 2024) and LiveCodeBench (LCB, execution split) (Jain et al., 2025) for code understanding.

**Constructing the Reasoning Dataset**   We sample reasoning traces using the hyperparameters recommended in the models' technical reports or HuggingFace repositories (see Appendix A). In total, we collect 24,444 reasoning traces across 12 LRMs on the five datasets.

**Training Setup**   For all experiments, we use Llama3.3-70B-Instruct (Dubey et al., 2024) as the annotator model because of its strong instruction-following capability and open-weight nature.

LOT training uses an 80-20 train-test split on MATH-500, GPQA-Diamond, CRUXEVAL, and LCB-execution, and a 75–25 split on AIME 24 & 25 due to its small size (60 questions). The taxonomy is initialized by comparing one reasoning pair and then expanded iteratively following Algorithm 1. A reasoning pair consists of traces from two LRMs given the same question. After each update to the taxonomy, the LLM annotates a batch of 40 additional pairs using the updated taxonomy. The logistic classifier is then re-trained on the updated embeddings.

**Anonymized Model Labels**   Model names often reveal attributes such as their scale, family, and domain specialization. To avoid biasing the LLM annotator, we assign each LRM a neutral codename (*e.g.,* "Omelet" for Phi-4-Reasoning-Plus) when training LOT.

### 4.1   DOES PARAMETER SCALE AFFECT A MODEL'S REASONING PROCESS?

We begin by examining how the reasoning patterns of LRMs vary with their parameter scales. Recent results show that the scaling law Snell et al. (2024) extends to LRMs, whose post-reasoning performance correlates with their size (Guo et al., 2025; Yang et al., 2025). Beyond task accuracy, we find that the "artificial brains" at different sizes also have systematic differences in their thinking.

We locate their differences by training LOTs to classify reasoning traces generated by Qwen3 models (Yang et al., 2025) of five smaller sizes (0.6B–14B parameters) against their largest variant, Qwen3-32B. Because the smaller Qwen3 models are distilled from Qwen3-32B, they form an ideal testbed for studying how parameter scale relates to reasoning behaviors.

**Classification Accuracy**   As Figure 1 shows, LOT achieves 80–93% accuracy across all datasets on classifying the traces of Qwen3-0.6B and Qwen3-32B, two models with the largest parameter gap. Incorporating frequency information (BoR) further improves accuracy by 3–14% over PoR encodings. However, as the parameter gap narrows, accuracy declines under both encodings, suggesting that the reasoning traces from models with closer scales are less distinguishable to LOT.

**Baselines**   We compare LOT against few-shot prompting (FSP) (Bai et al., 2022), VML (Xiao et al., 2025), and PoR/BoR built from a fixed, human-defined taxonomy (Gandhi et al., 2025). For the FSP baseline, each shot includes a pair of traces from the smaller model and Qwen3-32B for the same question. We sweep 1–15 shots per dataset and report the best $N$-shot result. VML uses the same $N$ as its update batch size, except on AIME, where a batch size of 2 was used. For the

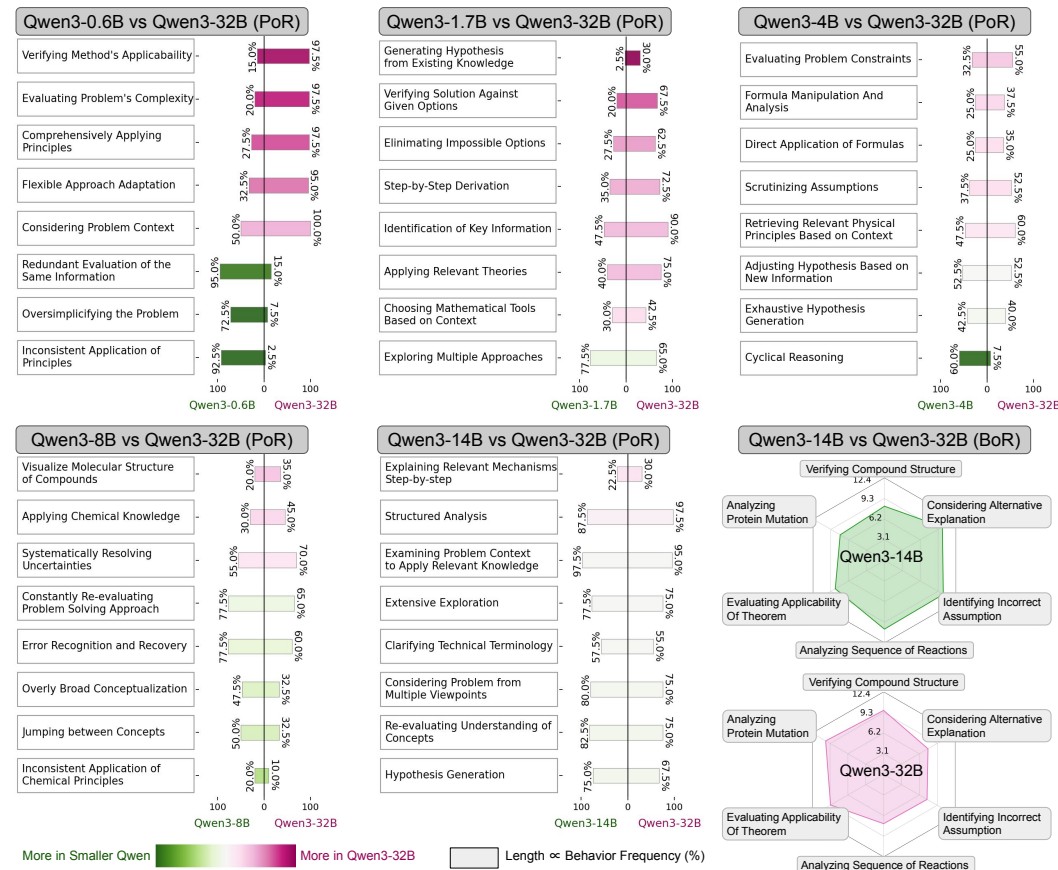

Figure 2: Reasoning differences between Qwen3-32B and its smaller variants on GPQA identified by LOT. Color indicates how often the reasoning trace $x$ with feature $c$ is from Qwen3-32B versus its smaller variant, $\mathbb{E}[\mathbf{1}_{\text{Qwen3-32B}(x)} \mid x_c = 1]$ on test split. Bar length, on each side, encodes the frequency of $c$ in the respective model's reasonings. Radar chart shows the averaged BoR encodings.

human-defined taxonomy, we annotate the entire reasoning dataset using the taxonomy and train a logistic regression classifier on the resulting embeddings. All methods use Llama3.3 for inference with same sampling hyperparameters. The prompts for FSP and VML are adapted from LOT's.

Across five datasets, PoR and BoR encodings of LOT outperform the baselines on almost every pairwise classification. The only exception is on MATH-500, where encodings using the fixed, human-defined taxonomy perform similarly to LOT on classifying Qwen3-4B/8B/14B versus 32B.

**Reasoning Differences** LOT also discovers reasoning differences that are not captured in the human-defined taxonomy. Figure 2 highlights some discriminative reasoning features between the smaller Qwen3 models and Qwen3-32B on the GPQA dataset (test split). In summary, **Qwen3-32B** more reliably recalls problem-relevant knowledge, checks the applicability of its chosen approaches against problem constraints and context, and executes step-by-step analyses without losing the thread. In contrast, smaller variants often redundantly evaluate the same information (*e.g.,* repeatedly stating the net field within a conductor is zero) which leads to circular reasoning. The LOT also observes smaller Qwen3 models, such as Qwen3-0.6B and Qwen3-8B, often fail to commit to a specific scientific theory or apply the wrong theory when solving the questions. As a result, they frequently switch hypotheses, shift concepts, and eventually confuse themselves.

Another interesting pattern is observed in Qwen3-8B and Qwen3-32B. Although both are text-only models, they sometimes "visualize" the molecular structure of compounds. Analyzing their reasoning traces with this annotation shows that both models write out the structural formula of the compound given in the problem statement to better examine its chemical bonds (see Appendix B).

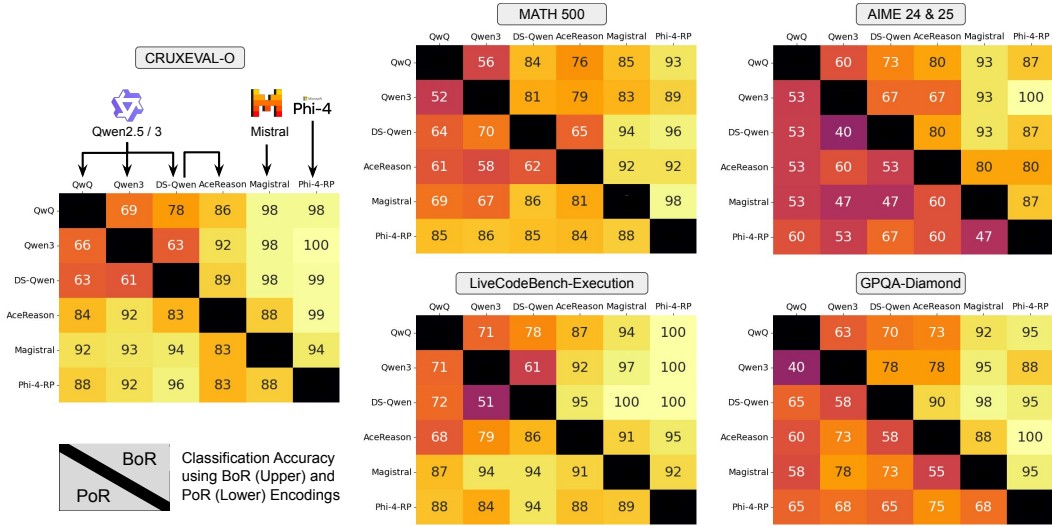

Figure 3: Accuracy in classifying reasoning traces of LRMs fine-tuned from different base models. Each cell shows test accuracy for the LRM in the row versus the LRM in the column, using PoR encodings (lower triangle) or BoR encodings (upper triangle). Arrows indicate fine-tuning relationships. Note that AceReason is RL fine-tuned from DS-Qwen whose base model is Qwen2.5.

## 4.2 CAN REASONING HABITS TELL US A LRM'S "ROOT"?

Beyond parameter scale, we compare models fine-tuned from different base model families and find notable differences in their thought patterns. Specifically, we apply LOT to six reasoning models trained on three base families: Qwen3-14B, QwQ-32B (Qwen, 2025), DS-Qwen-14B (Guo et al., 2025), and AceReason-Nemotron-14B (Chen et al., 2025), all based on Qwen; Magistral-Small based on Mistral (Rastogi et al., 2025b); and Phi-4-Reasoning-Plus based on Phi-4 (Abdin et al., 2025b). Except for Magistral-Small (24B) and QwQ-32B, all models have 14B parameters.

As shown in Figure 3, the accuracy in classifying traces from models with the same base (*e.g.,* DS-Qwen-14B and QwQ-32B) is lower, regardless of whether BoR (upper triangle of the heatmap) or PoR (lower triangle) encodings are used. This suggests that these models potentially exhibit similar reasoning patterns.

For longer reasoning traces on challenging benchmarks, GPQA and AIME, PoR encodings are insufficient to classify thought processes, even if they are from LRMs fine-tuned from different bases. Considering the frequency of reasoning features (BoR) improves accuracy, indicating that these LRMs may use a similar set of reasoning strategies on harder questions, but differ in how frequently they employ them.

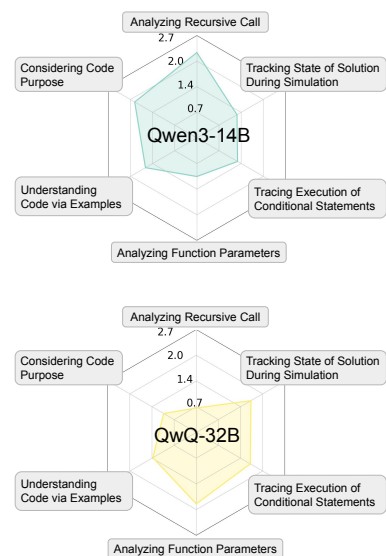

Figure 4: Qwen3-14B versus QwQ-32B on LCB-Execution. The chart shows the top-six most distinguishing features (not the entire LOT).

**Case study** What are the reasoning differences between these models? We examine the BoR encodings of Qwen3-14B and QwQ-32B's reasoning traces on LCB-Execution, which tests their understanding of Python code. Both models achieve high accuracy on this task ($\sim 98\%$), but they diverge in the number of steps used to understand function purposes and analyze recursive calls (see Figure 4). Two models also take different approaches in comprehending the provided code: Qwen3-14B, on average, spends more steps in simulating the code on various input–output examples, while QwQ-32B focuses more on analyzing input parameters and their contribution to the final output.

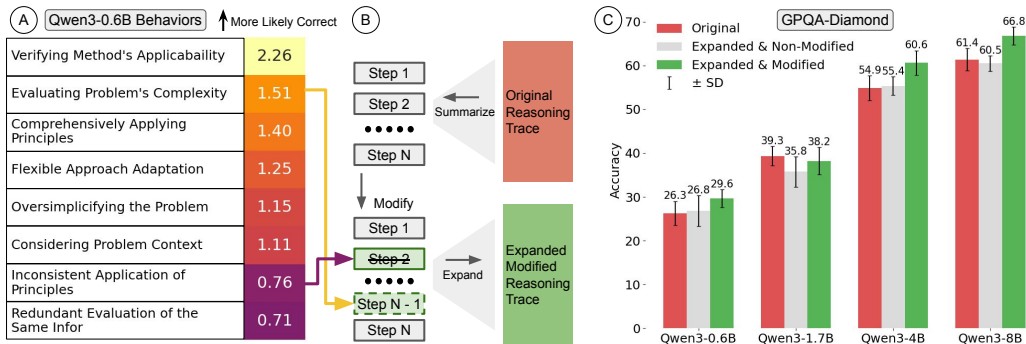

Figure 6: (A) shows the odds ratio for each reasoning feature $c$ in Qwen3-0.6B's reasoning on GPQA. (B) describes our intervention pipeline. (C) are the GPQA results after modifying the Qwen3 models' reasoning traces. The results are averaged across 10 runs.

### 4.3 Does task domain bring any inertia to LRM's reasoning habits?

Some models' reasoning capabilities are fine-tuned on a specific domain. Seed-Coder-8B-Reasoning, for example, is pretrained on a mixture of math and coding data, but its reasoning is fine-tuned solely on coding-related datasets. It is natural to ask how such a model reasons about problems outside its fine-tuning domain, such as math.

Applying LOT to classify Seed-Coder-8B-Reasoning's and Qwen3-8B's reasoning on MATH-500 reveals an intriguing difference: Seed-Coder sometimes borrows its coding-oriented reasoning style for mathematics. For most questions, Seed-Coder adopts a computational approach similar to Qwen3-8B. However, in 20% of cases, Seed-Coder goes further by implementing a Python function to solve the problem (subsection 4.3). Qwen3-8B also exhibits coding-based reasoning, but only in 2% of questions, specifically when the prompts contain Asymptote code describing diagrams. In those cases, Qwen3-8B simply interprets the graphic code without further coding-related actions. Seed-Coder, however, writes pseudocode, implements it in Python, and simulates its

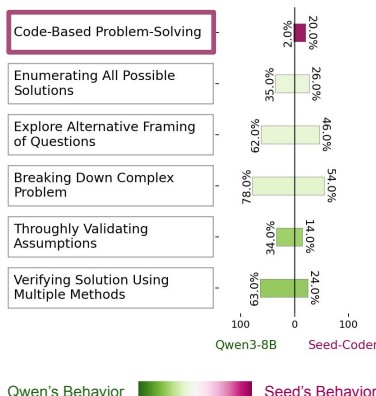

Figure 5: Qwen3 versus Seed-Coder-Reasoning on MATH 500. Colors, from green to purple, are proportional to $\mathbb{E}[\mathbf{1}_{\text{Seed}(x)} \mid x_c = 1]$.

execution to directly solve the problem, even when there is no code in questions. This suggests that fine-tuning on a specific domain may introduce a degree of "inertia" in an LRM's reasoning habits.

## 5 Connecting Reasoning Differences with Performance Gaps

Do discrepancies in models' reasoning habits help explain their performance differences? In this section, we demonstrate that the reasoning differences identified by LOT have both the correlational and causal links with models' performance gaps.

We utilize the LOTs trained on the Qwen3 models in subsection 4.1 and their annotations of reasoning traces on GPQA. For each feature $c$ that distinguishes a smaller Qwen3 model from its largest counterpart (Qwen3-32B), we compute the odds ratio $\frac{p(x \in \text{correct}|x_c=1)/p(x \in \text{wrong}|x_c=1)}{p(x \in \text{correct}|x_c=0)/p(x \in \text{wrong}|x_c=0)}$, which quantifies how much more likely a reasoning trace is to be correct when the feature $c$ appears versus not. Figure 6 reports these odds ratios for Qwen3-0.6B on GPQA. The results show the inconsistent application of scientific principles and redundant evaluation appear more often in Qwen3-0.6B's incorrect reasoning, while verifying a method's applicability is strongly associated with correct ones. Appendix J provides odds ratios for other Qwen3 models.

However, strong associations alone do not establish causality. Does the appearance of a reasoning behavior affect the correctness of a model's final answer? One way to test this counterfactual relation

is to instruct an LRM to perform reasoning behaviors more or less frequently, based on their odds ratios. Surprisingly, current LRMs, including the Qwen3 family, struggle to follow instructions about their *reasoning content*. In a baseline experiment (details in Appendix C), we prompt the LRMs to begin their reasoning with a specific sentence when solving a GPQA question. None of the open-source LRMs reliably generate the required sentence in their reasoning. In particular, the Qwen3 models often generate the sentence at the start of final (non-thinking) output, after completing their reasoning.

This observation motivates us to design an alternative intervention pipeline (Figure 6B). Given a model to be intervened on, we first instruct it to summarize its original reasoning, paragraph by paragraph, into a list of steps. Next, the model is prompted to edit this summary by adding or removing steps according to the correlation findings. Finally, the model iteratively re-expands the modified summary into a complete reasoning trace. All steps are conducted in the Qwen3 models' non-thinking mode[1]. We infer the final answer from the intervened model using the expanded reasoning as its thinking content. Why not instruct the Qwen3 models to edit their original reasoning directly? Summarization is necessary because some reasoning traces contain more than 20K tokens, and direct modification would exceed the 32K-token context window of the Qwen3 models.

Figure 6C shows that the intervention improves the accuracy of Qwen3-0.6B, Qwen3-4B, and Qwen3-8B on GPQA. To ensure that the gains came from the modifications rather than summarization alone, we evaluate re-expanded traces from unmodified summaries, and the comparisons confirm that the improvements from intervention are significant. The only exception is Qwen3-1.7B, whose performance drops significantly after summarization. As a result, the modified traces perform worse than the original traces, though they still outperform the unmodified expansions. This failure is potentially due to Qwen3-1.7B's poor instruction-following during reasoning (Appendix C).

## 6 LIMITATIONS AND FUTURE WORK

Our study has several limitations. First, we did not establish causal links between the meta-attributes of LRMs (*e.g.,* size) and their reasoning patterns. Most LRMs we compared do not fully open source their training recipes and may differ in several meta-attributes beyond those we focused on. Comparing models trained under controlled conditions would allow us to make stronger causal connections, but that requires significant compute inaccessible to us.

Second, our reasoning taxonomy is sampled from an LLM and may thus vary with random seeds. In Appendix D, we assess the consistency of taxonomies generated from five different seeds. We find that, after a sufficient number of iterations, the five taxonomies converge to a similar set of features.

Since LOT is optimized for classification, we make no guarantee that a trained LOT will describe the complete set of reasoning differences between LRMs. For example, if two LRMs consistently differ in multiple reasoning styles, finding any subset of them will lead the LOT's training to convergence.

Finally, the approaches for modifying the LRMs' reasoning styles are worth future study. We described a test-time method in section 5, but future work may explore how to leverage the identified reasoning differences in training such as using them to select fine-tuning data or incorporating them as processed reward for reinforcement learning. Meanwhile, our work used Llama3.3-70B-Instruct in all experiments. The effects of the LLM annotator on LOT's performance remain underexplored.

## 7 CONCLUSION

This work introduced LOT, a classification method that produces human-readable taxonomies of LRMs' reasoning differences, accurately distinguishing their thought processes. We apply LOT to compare the reasoning behaviors of 12 open-source LRMs, and it achieves higher classification accuracy than a predefined reasoning taxonomy and automatic prompt engineer approaches. Beyond classification, LOT enables direct interpretation of how LRMs reason differently. Through intervention experiments, we show that certain reasoning differences contribute to performance gaps between models. In particular, we improve GPQA performance by 3.3–5.7% by modifying the reasoning behaviors of smaller Qwen models during test time.

---

[1]Qwen3 models are trained with thinking control that allows them to generate answers without thinking.

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

# A  CONSTRUCTING THE REASONING DATASET

## A.1  HYPERPARAMETER USED IN SAMPLING REASONING OUTPUTS FROM LRMs

Table 1 shows the sampling hyperparameters we use to generate reasoning traces from each LRM. For each model, the same hyperparameters are applied across all datasets.

Seed-Coder-8B-Reasoning (Seed et al., 2025)'s technical report and HuggingFace (HF) repository do not specify the sampling hyperparameters used in the evaluation. However, the technical report states that a temperature of 0.6 is used when training Seed-Coder for reasoning. For Top-p and Top-k, we use the most common numbers observed in the other LRMs.

Table 1: Sampling hyperparameters used for each LRM.

| Models | Temp | Top-p | Top-k | Source |
|---|---|---|---|---|
| Qwen3 Family | 0.6 | 0.95 | 20 | Paper (Yang et al., 2025) |
| AceReason-Nemotron-14B | 0.6 | 0.95 | 50 | Paper (Chen et al., 2025) |
| DeepSeek-R1-Distill-Qwen-14B | 0.6 | 0.95 | 50 | Paper (Guo et al., 2025) |
| QwQ-32B | 0.6 | 0.95 | 20 | HF Repo (Qwen, 2025) |
| Magistral-Small | 0.7 | 0.95 | 50 | HF Repo (Rastogi et al., 2025a) |
| Phi-4-reasoning-plus | 0.8 | 0.95 | 50 | HF Repo (Abdin et al., 2025a) |
| Seed-Coder-8B-Reasoning | 0.6 | 0.95 | 50 | — |

## A.2  PROMPT TEMPLATES USED IN SAMPLING REASONING OUTPUT

We use prompt templates in Figure 7 when sampling reasoning traces and answers from the LRMs. The prompt template for math datasets is adopted from the `promptbase` library. AceReason-Nemotron, Qwen3, Magistral, and DeepSeek also recommend using "$
boxed" to format final outputs on math questions (mentioned in their HuggingFace repository). The prompt template for GPQA-Diamond is adopted from Zhou et al. (2025). For CRUXEVAL and the LiveCodeBench execution split, we use the prompt template provided in the original CRUXEVAL paper (Gu et al., 2024).

---

**Sampling Prompt Used for MATH-500 and AIME 24 & 25**

{Question statement}

Please end your solution with Answer: $\\boxed{number}$ where number is the numerical answer without unit.

---

**Sampling Prompt Used for GPQA-Diamond**

{Question statement}
A: {Choice}
B: {Choice}
C: {Choice}
D: {Choice}

Please reason step by step, and put your final answer within \\boxed{}.
Please only provide the letter of the answer in the box.

---

**Sampling Prompt Used for CRUXEVAL and LCB**

Based on the given Python code, which may contain errors, complete the assert statement with the output when executing the code on the given test case. Do not output any extra information, even if the function is incorrect or incomplete.

[PYTHON]
{function provided by the dataset}
[/PYTHON]

assert {function_name}({example_input}) == ??

In your final output, surround your answer with no additional words, with [ANSWER] and [/ANSWER] tags. Your answer should be [ANSWER] [Expected outputs] [/ANSWER]

---

Figure 7: Prompt templates used in sampling reasoning responses to each dataset.

# B ADDITIONAL QUALITATIVE ANALYSES OF GENERATED TAXONOMIES

## B.1 REPEATED VERIFICATION OF OUTPUT FORMAT

We observe an unexpected reasoning behavior in Phi-4-RP when comparing its BoR encodings with those of AceReason on CRUXEVAL: Phi-4-RP excessively checks the format of its final answer to ensure compliance with the instructions. In CRUXEVAL, models are asked to predict a Python function's output for a given input. The questions also ask the model to enclose its predictions by "[ANSWER]" and "[/ANSWER]" with no extra words. Although both models perform similar on this task ($\sim 84\%$ accuracy), Phi-4-PR spends significantly more steps verifying its adherence to the required format. Below is a reasoning trace from Phi-4-PR that exemplifies this.

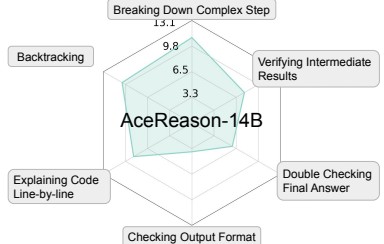

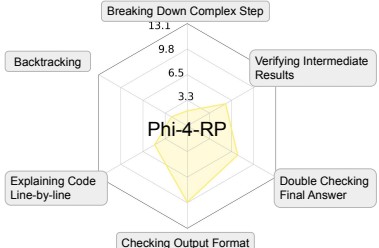

> [Previous reasoning ......]
>
> Thus answer: 9.
>
> But wait: The question says "complete the assert statement with the output when executing the code on the given test case". And then it says "Your answer should be [ANSWER] [Expected outputs] [/ANSWER]" with no extra words.
>
> Thus answer: [ANSWER]9[/ANSWER]. But careful: The instructions say "Do not output any extra information, even if the function is incorrect or incomplete." So answer should be exactly that.
>
> I'll produce the answer in the specified format: [ANSWER]9[/ANSWER]. I need to output answer with no extra words. So answer: [ANSWER]9[/ANSWER].

Figure 8: AceReason versus Phi-4-RP on CRUXEVAL (code understanding). Radar chart shows the averaged BoR encodings. Note that the chart highlights the reasoning features with the largest differences. It is not the complete set of features identified by LOT.

While this habit may appear unusual from a human reasoning perspective, it is important for LRMs, whose correct answers are rewarded during training only if they can be parsed by the outcome reward function. Outcome reward functions extract the final answer using regular expressions that require a strict output format. Phi-4-RP uses more steps to ensure its solutions can be properly extracted.

## B.2 VISUALIZING MOLECULAR STRUCTURE OF COMPOUNDS

In subsection 4.1, we mention that Qwen3-8B and Qwen3-32B (text-only models) attempt to "visualize" the molecular structure of the compounds given in the problem. Examining their reasoning traces with the annotation of this behavior, we find that the two models often convert the compounds originally expressed in their IUPAC names into structural formulas that more explicitly describe the chemical bonds within molecules (see Figure 9 for examples).

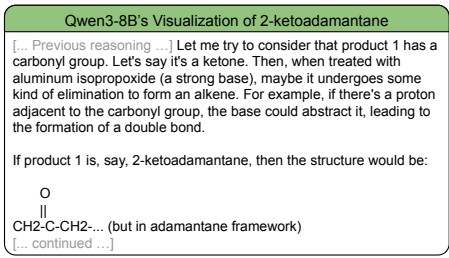

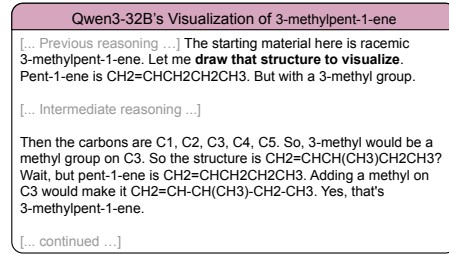

Figure 9: Examples of Qwen3-8B and Qwen3-32B "visualize" the molecular structure of compounds by writing their structural formula.

### B.3 CODE-BASED REASONING

In subsection 4.1, we apply LOT to classify Qwen3-8B's and Seed-Coder-8B-Reasoning's reasoning on MATH-500. One of the reasoning features in LOT suggests that Seed-Coder occasionally uses a coding approach to solve math problems, where the model writes pseudocode, provides its Python implementation, and simulates execution. We provide an example of this phenomenon in Figure 10.

Qwen3-8B exhibits code-based reasoning in only 2% of the questions of which the prompts contain Asymptote code describing a diagram. In those questions, Qwen3-8B simply parses information from the graphic code that is necessary for problem solving without taking further coding-related actions (see Figure 11 for an example).

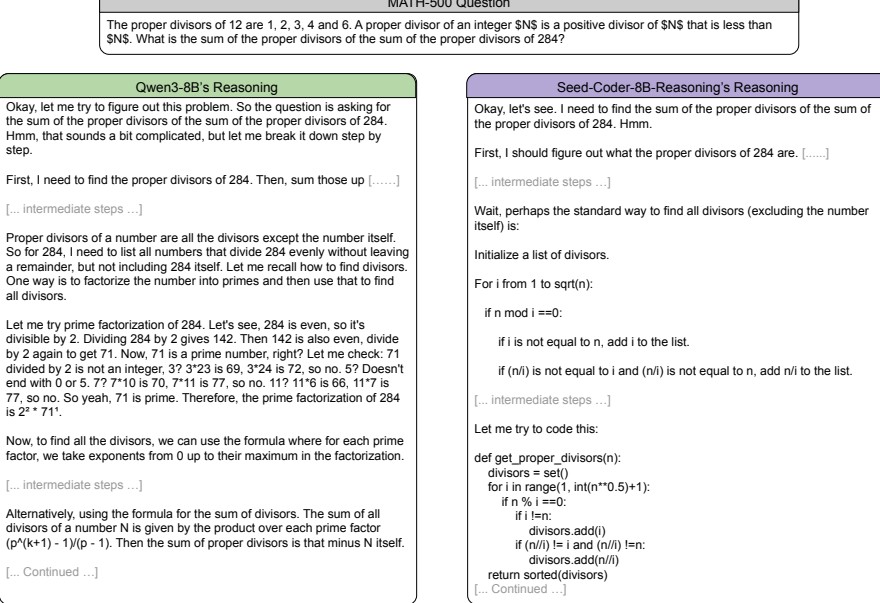

Figure 10: Qwen3-8B approaches the problem with its math knowledge and obtains the results by direct calculations. Seed-Coder-8B-Reasoning uses a similar math concept (proper divisors of 284 are numbers that divide 284 with no residual) but outlines it in pseudocode and implements it in Python.

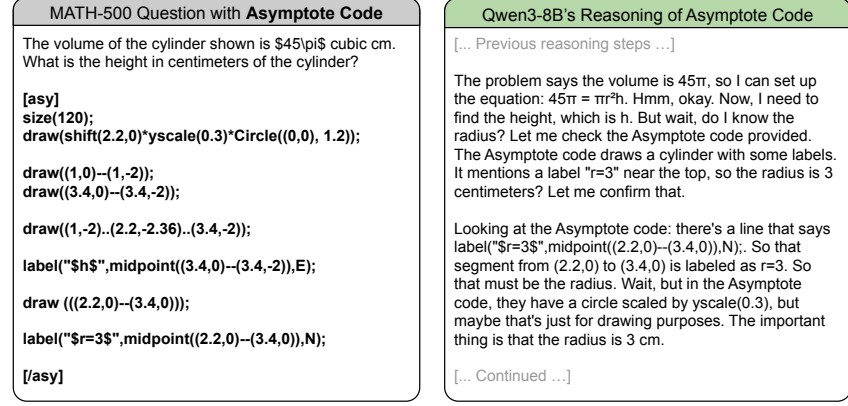

Figure 11: Qwen3-8B's code-based reasoning only occurs when the question prompts contain Asymptote code that describes diagram necessary for solving the problem.

## C    INSTRUCTION FOLLOWING DURING REASONING

In section 5, we describe an intervention experiment in which we modify specific reasoning behaviors in model traces and measure the resulting changes in correctness.

One way to implement such modifications is to prompt LRMs to perform certain behaviors more or less frequently when solving a question. To test the feasibility of this approach, as a minimum check, we instruct the LRMs to generate the sentence *"I am a large language model."* at the *beginning of their thinking* while solving questions from GPQA. Although simple, this test can reveal whether an LRM can insert designated content at a specified location within its reasoning process.

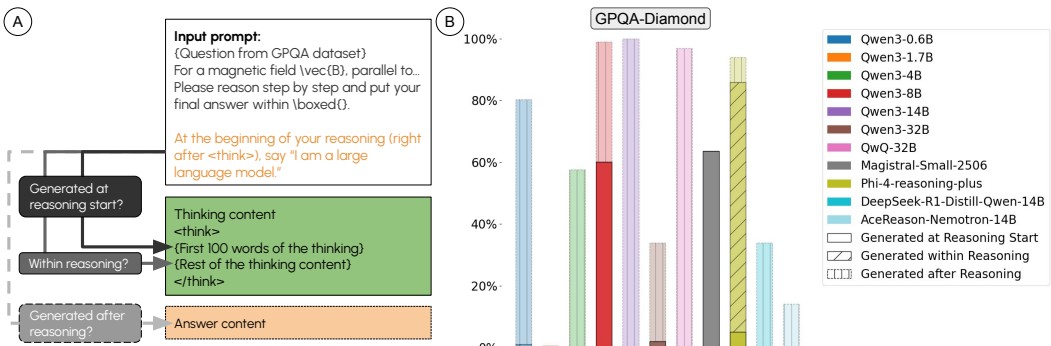

Figure 12: Existing open-source LRMs are incapable of following instructing about their reasoning content. (B) reports the percentage of responses in which an LRM generated the target sentence within the first 100 reasoning words (solid bar), elsewhere in the reasoning but not at the beginning (hatched bar), or only after the reasoning (hatched bar with dashed border). To steer an LRM's reasoning through prompting, we need the LRM to reliably follow our instruction (a high solid bar).

Surprisingly, almost all LRMs in our study fail this task (see Figure 12). Qwen3-8B and Magistral-Small are the only models that generate the sentence at the start of their reasoning with probabilities slightly above 50% on 198 GPQA questions. Phi-4-RP produces the sentence in roughly 90% of cases, but most often at the end of its reasoning rather than at the beginning. Other models, such as Qwen3-14B and QwQ-32B, produce the sentence at the start of their non-reasoning content instead. Among all models, Qwen3-1.7B performs the worst, almost never producing the required content in its entire outputs.

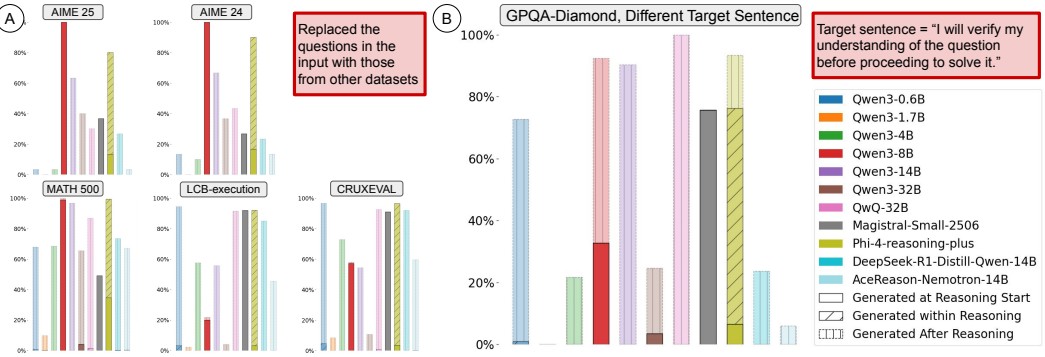

Figure 13: Instruction adherence of LRMs when given questions from other datasets in the input (A) or when instructed to generate a different sentence (B).

One may ask whether the failure comes from the choice of the target sentence. Indeed, *"I am a large language model"* is a factual statement but unrelated to the rest of the thinking process. To test this, we repeat the above intervention experiment with different target sentences, such as *"I will*

*verify my understanding of the question before proceeding to solve it."* which is directly related to reasoning. Nonetheless, our results in Figure 13B show that the same failures persists.

Furthermore, Figure 13A shows that models failing completely on GPQA also fail across other benchmarks. For those that do follow instructions on GPQA, their performance is sensitive to the question in the context. Magistral, for example, adheres to instructions better on coding datasets but worse on math benchmarks. In contrast, Qwen3-8B adheres to instructions better on the math-related datasets.

While we find these findings intriguing, instruction-following is not the main focus of this work. We hope our observations will motivate future work on understanding and improving instruction adherence in reasoning generation.

## D    STABILITY OF TAXONOMY GENERATION

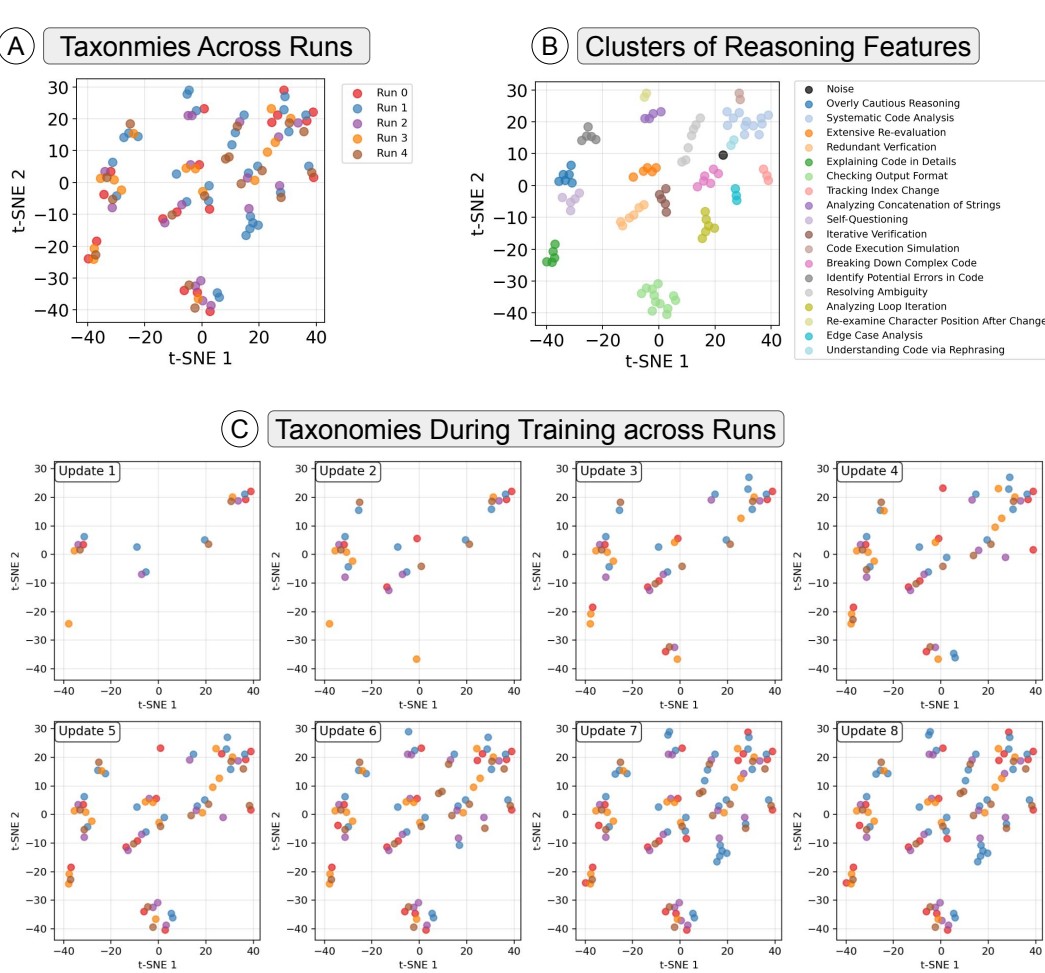

Figure 14: t-SNE visualization of reasoning features in LOTs generated using 5 random seeds on classifying the DS-Qwen and Phi-4-PR's reasonings on CRUXEVAL. Each dot corresponds to the `gtr-t5-base` sentence embedding of a reasoning feature's name and definition generated by Llama3.3.

LOT uses an LLM, a probabilistic model, to compare reasoning traces and generate the names and definitions of reasoning features that are later used in classifying LRMs' outputs. A natural question is whether this feature-generation process is stable: if we train LOT multiple times, do we obtain significantly different taxonomies each time?

To answer this question, we train LOT five times to classify DS-Qwen and Phi-4-RP's reasoning on the CRUXEVAL dataset, each with a **unique random seed**. All trainings use Llama3.3-70B-Instruct as the inference model with the same sampling hyperparameters.

Across the five runs, LOT produces 93 reasoning features, with an average of 18 features per run. We convert each feature's name and definition into embeddings using `gtr-t5-base`. Figure 14A shows a t-SNE visualization of the embeddings. Applying DBSCAN to the embeddings yields 17 clusters shown in Figure 14B. We manually check the reasoning features in each cluster and annotate their themes on the right of Figure 14B.

An important observation is that the reasoning taxonomies generated across multiple runs cover almost the same thematic set. Most of the clusters contain reasoning features generated in at least four of the five runs. Three clusters include features from three runs, two clusters include features from only two runs, and only one cluster includes the feature from a single run.

We additionally plot the evolution of taxonomies from different runs during training in Figure 14C. In the first five updates, many reasoning features appear in only 1 or 2 runs. However, these features are gradually discovered by other runs in subsequent updates. After sixth update, most of the reasoning features are discovered in 4 out of 5 runs. Test set classification accuracies from five runs are also similar, with an average of 97.2% and a standard deviation of 2.1%.

# E  COMPARING BASELINE METHODS ON CLASSIFYING LRMS FINE-TUNED FROM DIFFERENT BASE MODELS

In subsection 4.1, we show that PoR/BoR encodings built from LOT outperform all baselines when classifying reasoning traces generated by Qwen3 models of different parameter scales. We find that this advantage also extends to models fine-tuned from different base model families.

Table 2 compares the accuracy of all baselines and LOT when classifying reasoning traces from AceReason-Nemotron-14B against each of the other models tested in subsection 4.2. Across all model pairs and datasets, BoR with LOT consistently achieves the highest accuracy. PoR also outperforms VML and most few-shot prompting (FSP) settings up to 10-shot.

Note that accuracy is not available for some FSP settings because the reasoning traces on those datasets are too long, causing $N$-shot examples to exceed the 128K context window of Llama3.3. For instance, on the AIME dataset, a single reasoning trace contains 16K tokens on average, and a shot consists of one trace from each model. The context window of Llama3.3 is nearly full with three shots plus the traces to be classified. Moreover, prior work (Agarwal et al., 2024; Tang et al., 2025) observes that the performance of FSP decreases after a certain number of shots. We observe a similar trend: on MATH-500, CRUXEVAL, and LCB-execution, the accuracy of FSP typically declines or plateaus after 5-shot examples.

Table 2: Classification accuracy of baseline methods and LOT. PoR(Fixed) and BoR(Fixed) are encodings generated from a fixed, human-predefined taxonomy. "—" in the few-shot settings indicates that the $N$-shot input exceeds the 128K-token context window of the LLaMa3.3 model.

| | | | | | | | AceReason-Nemotron-14B versus QwQ-32B | | | |
| Shots | 1 | 3 | 5 | 7 | 10 | VML | PoR (Fixed) | **PoR (LOT)** | BoR (Fixed) | **BoR (LOT)** |
|---|---|---|---|---|---|---|---|---|---|---|
| GPQA | 60% | 50% | 45% | — | — | 43% | 50% | 60% | 70% | **73%** |
| AIME 24/25 | 53% | — | — | — | — | 40% | 60% | 53% | 67% | **80%** |
| MATH 500 | 53% | 55% | 53% | 54% | 54% | 42% | 57% | 61% | 69% | **76%** |
| CRUX | 47% | 49% | 50% | 54% | 54% | 61% | 58% | 84% | 82% | **86%** |
| LCB | 48% | 49% | 53% | 51% | 49% | 50% | 57% | 68% | 73% | **87%** |

| | | | | | | | AceReason-Nemotron-14B versus Qwen3-14B | | | |
| Shots | 1 | 3 | 5 | 7 | 10 | VML | PoR (Fixed) | **PoR (LOT)** | BoR (Fixed) | **BoR (LOT)** |
|---|---|---|---|---|---|---|---|---|---|---|
| GPQA | 50% | 50% | 50% | 43% | — | 50% | 53% | 73% | 75% | **78%** |
| AIME 24/25 | 53% | 53% | — | — | — | 33% | 47% | 60% | 60% | **67%** |
| MATH 500 | 56% | 57% | 55% | 56% | 57% | 40% | 55% | 58% | 62% | **79%** |
| CRUX | 68% | 71% | 68% | 69% | 59% | 77% | 76% | 92% | 90% | **92%** |
| LCB | 69% | 68% | 80% | 75% | 65% | 48% | 73% | 79% | 88% | **92%** |

| | | | | | | | AceReason-Nemotron-14B versus DeepSeek-R1-Distill-Qwen-14B | | | |
| Shots | 1 | 3 | 5 | 7 | 10 | VML | PoR (Fixed) | **PoR (LOT)** | BoR (Fixed) | **BoR (LOT)** |
|---|---|---|---|---|---|---|---|---|---|---|
| GPQA | 48% | 46% | — | — | — | 45% | 58% | 58% | 87% | **90%** |
| AIME 24/25 | 53% | 54% | — | — | — | 47% | 53% | 53% | 67% | **80%** |
| MATH 500 | 55% | 57% | 61% | 55% | 59% | 43% | 54% | 62% | 54% | **65%** |
| CRUX | 49% | 59% | 60% | 53% | 53% | 78% | 70% | 83% | 88% | **89%** |
| LCB | 61% | 62% | 56% | 65% | 74% | 69% | 66% | 86% | 89% | **95%** |

| | | | | | | | AceReason-Nemotron-14B versus Magistral-Small | | | |
| Shots | 1 | 3 | 5 | 7 | 10 | VML | PoR (Fixed) | **PoR (LOT)** | BoR (Fixed) | **BoR (LOT)** |
|---|---|---|---|---|---|---|---|---|---|---|
| GPQA | 50% | 55% | 50% | — | — | 50% | 50% | 55% | 75% | **88%** |
| AIME 24/25 | 53% | 40% | — | — | — | 33% | 53% | 60% | 67% | **80%** |
| MATH 500 | 59% | 60% | 62% | 53% | — | 50% | 58% | 81% | 91% | **92%** |
| CRUX | 76% | 84% | 78% | 83% | — | 31% | 63% | 83% | 76% | **88%** |
| LCB | 45% | 49% | 63% | 48% | 47% | 56% | 55% | 91% | 87% | **91%** |

| | | | | | | | AceReason-Nemotron-14B versus Phi-4-reasoning-plus | | | |
| Shots | 1 | 3 | 5 | 7 | 10 | VML | PoR (Fixed) | **PoR (LOT)** | BoR (Fixed) | **BoR (LOT)** |
|---|---|---|---|---|---|---|---|---|---|---|
| GPQA | 63% | 68% | 60% | — | — | 50% | 50% | 75% | 93% | **100%** |
| AIME 24/25 | 53% | — | — | — | — | 47% | 40% | 60% | 73% | **80%** |
| MATH 500 | 87% | 88% | 85% | 85% | 86% | 52% | 69% | 84% | 84% | **92%** |
| CRUX | 76% | 95% | 97% | 95% | 95% | 81% | 77% | 83% | 89% | **99%** |
| LCB | 53% | 57% | 52% | 51% | 63% | 49% | 86% | 88% | 65% | **95%** |

# F   PROMPTS FOR LOT ANNOTATION AND UPDATE

We provide the following prompt templates to Llama3.3 for generating taxonomies and annotations. We sample its outputs using a temperature of 0.6, a top-p of 0.95, and a top-k of 50. All steps of LOT use the same hyperparameters. We use the official checkpoint of Llama3.3 provided on its HuggingFace repository.

---

## Prompt for Generating BoR Annotations with LOT

You are an expert in qualitative research and grounded theory, and you are good at annotating the reasoning behaviors of language models' generated reasoning using a taxonomy of reasoning behaviors.

You will be given a language model's reasoning trace (OUTPUT A or OUTPUT B) toward a question.

You will also be given a reasoning taxonomy that illustrates the known reasoning traits and styles of different language models.

Your task is to annotate the reasoning behaviors (in the taxonomy) that appeared in the given reasoning trace based on their definitions in the given taxonomy.

Think step by step. You should annotate the given OUTPUT using the reasoning behaviors in the taxonomy following the provided definitions.

You don't need to use every reasoning behavior in the reasoning taxonomy in your annotation. It's possible some reasoning behaviors do not occur in the given output.

On the other hand, the same reasoning behaviors may appear multiple times at different places in the reasoning output.

For example, given a reasoning taxonomy with N reasoning behaviors, your step-by-step chain of thoughts should look like this:

------
[Annotate the reasoning OUTPUT with the given taxonomy]: [Beginning of the OUTPUT A or B] [Summarize the first sentence + Behavior Name for first sentence] [Summarize the second sentence + Behavior Name for second sentence] [continue for the rest of the sentences] ... [Summarize the last sentence + Behavior Name for last sentence] [End of the OUTPUT A or B] {YOU MUST ANNOTATE THE WHOLE REASONING OUTPUT A or B} {If OUTPUT A or B has multiple paragraphs, annotate the sentences in all paragraphs}

{Your should annotate the OUTPUT sentence by sentence. For each sentence, represent it with one of the reasoning behavior if applicable. Use [Not in Taxonomy] for behaviors not described by the given taxonomy. Don't be lazy even if the OUTPUT is long!}
------

Make sure your output chain of thoughts follows this format exactly.

You must annotate the whole reasoning OUTPUT given to you.

Below is the reasoning taxonomy that you will use for the annotation,
{Reasoning taxonomy}

---

Figure 15: Instruction for generating bag of reasoning (BoR) annotation of a given reasoning trace.

## Prompt for Updating LOT when using BoR

You are an expert in qualitative research and grounded theory, and you are good at distinguishing the reasoning behaviors of different language models.

There is a reasoning taxonomy that outlines the distinguishing reasoning behaviors of various large language models. Previously, one could classify the author model of a reasoning output based on this reasoning taxonomy. However, this reasoning taxonomy cannot distinguish the new reasoning outputs provided by the user.

Your task is to identify missing distinguishing reasoning behaviors and add them to the reasoning taxonomy so that one can accurately classify these new reasoning outputs. Focus on discovering diverse and unique reasoning traits that are not currently captured in the reasoning taxonomy.

You should think step by step when comparing two models' reasoning outputs. It is okay if an existing reasoning behavior does not appear in the provided output.

If there are distinguishing differences in reasoning behaviors, but they are not included in the reasoning taxonomy, you should add a new reasoning behavior for each of those differences in the reasoning taxonomy.

When adding the new reasoning behavior, you should provide a short name of the reasoning behavior with its detailed definition, such as [Reasoning behavior name]: [What this reasoning trait is about] [Example of this behavior quoted from the given outputs]. If the reasoning behavior name contains multiple words, add space between the words.

[Example of this behavior] can be a direct quote. Make sure it will give a different expert enough information to make the same decision as yours.

Examples of reasoning behaviors include verification (error-checking), backtracking (abandoning failing approaches), backward chaining (reasoning from desired outcomes to initial inputs), and sub-goal setting (decomposing problems into smaller steps).

Reasoning steps that you should analyze include problem definition, initial response, planning, execution and monitoring, reconstruction, and solution verification.

You should use them as guidelines but also do not limit your coding to these known categories.

Your output step-by-step chain of thoughts should look like this: {Chain of thought format}

Make sure you follow the exact format above when giving the added reasoning behavior. Write the reasoning behavior name, reasoning behavior definition, and example in the same line (one line).

For the added reasoning behavior, think creatively. The added reasoning behaviors must separate two given outputs---that is it must occur significantly more in one of the outputs or only occur in one reasoning output. For example, it occurs in one of the outputs 7 times but only 3 times in other output. Or, it occurs in one of the reasoning outputs 1 time but not at all in other output.

Moreover, it should be different from the existing ones. Do not add reasoning behaviors that are similar to the existing ones in the reasoning taxonomy below in your Final output.

Below is the reasoning taxonomy you could use for the annotation,
{Reasoning Taxonomy}

Figure 16: Instruction for updating the taxonomies used in making BoR annotations. This instruction is similar to the update instruction used for PoR while a key difference is that the BoR instruction asks the LLM to extract reasoning behaviors that are either uniquely presented in one model's output or **appear more** in one of the outputs.

---

### Chain-of-thought Format for Updating LOT when using BoR

------
[Start by comparing the annotated the reasoning traces]
[Annotate the additional distinguishing reasoning behaviors in OUTPUT A]: [Summarize the first corresponding sentence + New Distinguishing Behavior Name for that sentence] [continue for the rest of the sentences (if any)] ... [Summarize the last corresponding sentence + New Distinguishing Behavior Name for last sentence] [End of the OUTPUT A] {If OUTPUT A has multiple paragraphs, annotate the sentences in all paragraphs}

[Annotate the additional distinguishing reasoning behaviors in OUTPUT B]: [Summarize the first corresponding sentence + New Distinguishing Behavior Name for that sentence] [continue for the rest of the sentences (if any)] ... [Summarize the last corresponding sentence + New Distinguishing Behavior Name for last sentence] [End of the OUTPUT B] {If OUTPUT B has multiple paragraphs, annotate the sentences in all paragraphs}

Now, I will summarize my new annotation for each OUTPUT, and then count number of behaviors that occurred in each OUTPUT.

{If you observe the distinguishing reasoning behaviors that are not in the reasoning taxonomy}
{Add new distinguishing reasoning behaviors}
### [New distinguishing reasoning behavior's name]
### [Definition of this reasoning behavior (reasoning behavior); a quote or detailed summarization of this behavior]
### [Whether this new reasoning behavior occurs in OUTPUT A: Either "This reasoning behavior is observed in OUTPUT A." or "This reasoning behavior is not observed in OUTPUT A." DO NOT USE ANY OTHER EXPRESSIONS OR ADD OTHER DETAILS.]
### [How many times this behavior occurs in OUTPUT A: "Count in OUTPUT A: {number}". DO NOT USE ANY OTHER EXPRESSIONS OR ADD OTHER DETAILS.]
### [Whether this new reasoning behavior occurs in OUTPUT B: Either "This reasoning behavior is observed in OUTPUT B." or "This reasoning behavior is not observed in OUTPUT B." DO NOT USE ANY OTHER EXPRESSIONS OR ADD OTHER DETAILS.]
### [How many times this behavior occurs in OUTPUT B: "Count in OUTPUT B: {number}". DO NOT USE ANY OTHER EXPRESSIONS OR ADD OTHER DETAILS.]
### [Is this reasoning behavior and its definition really different from the existing reasoning behavior above? If not, then you shouldn't include this reasoning behavior in the Added Rule section of your final output!]
...{Repeat for the other new reasoning behaviors}...

[Are new distinguishing reasoning behaviors above truly different from any of the existing ones in the system message? Again, you don't need to add a behavior unless it's truly different from known ones. Think step by step.]

Final output:
Added:
[Added distinguishing reasoning behavior name]: [Detailed reasoning behavior definition (reasoning behavior)] [Example of this behavior quoted from the given outputs or a detailed summarization of this behavior]
...{Repeat for the other new reasoning behaviors}...
------

Figure 17: The example chain-of-thought format that we provided to the LLM annotator when instructing it to generate the updates to the taxonomy.

## Prompt for Generating PoR Annotations with LOT

You are an expert in qualitative research and grounded theory, and you are good at annotating the reasoning behaviors of language model's generated reasoning using a taxonomy of reasoning behaviors.

You will be given two models' reasoning traces toward a question.

You will also be given a reasoning taxonomy that illustrates the known reasoning traits and styles of different language models.

Your task is to annotate the presence of reasoning behaviors in the given reasoning OUTPUTs based on the reasoning behaviors' definitions in the taxonomy. The presence of a reasoning behavior can inform us which language model generates the reasoning OUTPUT.

Think step by step. You should cite every reasoning behavior from the reasoning taxonomy, and explain whether the associated reasoning behavior described is observed in any outputs.

For example, given a reasoning taxonomy with N reasoning behaviors, your output step-by-step chain of thoughts should look like this:

------
{Compare two reasoning OUTPUTs side-by-side. Write a detailed summary of your analysis}

{Annotate the occurrence of reasoning behaviors in each OUTPUT following their definition in the reasoning taxonomy.}
### [Reasoning Behavior name 1]
### [Definition of this reasoning behavior; reasoning with the given output; which output shows this reasoning behavior (with a quote)]
### [Whether this reasoning behavior occurs in OUTPUT A: Either "This reasoning behavior is observed in OUTPUT A." or "This reasoning behavior is not observed in OUTPUT A." DO NOT USE ANY OTHER EXPRESSIONS OR ADD OTHER DETAILS.]
### [Whether this reasoning behavior occurs in OUTPUT B: Either "This reasoning behavior is observed in OUTPUT B." or "This reasoning behavior is not observed in OUTPUT B." DO NOT USE ANY OTHER EXPRESSIONS OR ADD OTHER DETAILS.]
...
### [Reasoning Behavior name N]
### [Definition of this reasoning behavior; reasoning with the given output; which output shows this reasoning behavior (with a quote)]
### [Whether this reasoning behavior occurs in OUTPUT A: Either "This reasoning behavior is observed in OUTPUT A." or "This reasoning behavior is not observed in OUTPUT A." DO NOT USE ANY OTHER EXPRESSIONS OR ADD OTHER DETAILS.]
### [Whether this reasoning behavior occurs in OUTPUT B: Either "This reasoning behavior is observed in OUTPUT B." or "This reasoning behavior is not observed in OUTPUT B." DO NOT USE ANY OTHER EXPRESSIONS OR ADD OTHER DETAILS.]
------

Think step by step.

Make sure your output chain of thoughts follow this format exactly (including the hashtags ###).

Below is the reasoning taxonomy that you will use for the annotation,
{Reasoning taxonomy}

Figure 18: Instruction for generating presence of reasoning (PoR) annotation of given reasoning traces.

## Prompt for Generating PoR Annotations with LOT

You are an expert in qualitative research and grounded theory, and you are good at distinguishing the reasoning behaviors of different language models.

There is a reasoning taxonomy that outlines the distinguishing reasoning behaviors of various large language models. Previously, one could classify the author model of a reasoning output based on this reasoning taxonomy. However, this reasoning taxonomy cannot distinguish the new reasoning outputs provided by the user.

Your task is to identify missing reasoning behaviors and add them to the reasoning taxonomy so the taxonomy can accurately classify the source language models that generate these new reasoning outputs. Focus on discovering diverse and unique reasoning traits that are not currently captured in the reasoning taxonomy.

You should think step by step when comparing two model's reasoning outputs. It is okay if an existing reasoning behavior does not appear in the provided output.

If there are distinguishing differences in reasoning behaviors, but they are not included in the taxonomy, you should add each of those reasoning differences in the taxonomy.

When adding the new reasoning behavior, you should provide a short name of the reasoning behavior with its detailed definition, such as [Reasoning behavior name]: [What this reasoning trait is about] [Example of this behavior quoted from the given outputs]. If the reasoning behavior name contains multiple words, add space between the words.

[Example of this behavior] can be a direct quote. Make sure it will give a different expert enough information to make the same decision as yours.

Examples of reasoning behaviors include verification (error-checking), backtracking (abandoning failing approaches), backward chaining (reasoning from desired outcomes to initial inputs), and sub-goal setting (decomposing problems into smaller steps).

Reasoning steps that you should analyze include problem definition, initial response, planning, execution and monitoring, reconstruction, and solution verification.

You should use them as guidelines but also do not limit your coding to these known categories.

For example, your step-by-step chain-of-thoughts should look like this: {Chain of thought format}

Make sure you follow the exact format above when giving the added reasoning behavior. Write the reasoning behavior name, reasoning behavior definition, and example in the same line.

You could add or update multiple reasoning behaviors to the reasoning taxonomy. It's possible more than one reasoning behavior needs to be added or changed. For the added reasoning behavior, think creatively.

Below is the reasoning taxonomy you could use for the annotation,

Figure 19: Instruction for updating the taxonomies used in making PoR annotations. This instruction is similar to the update instruction used for BoR. One difference is that the PoR instruction asks the LLM to annotate reasoning behaviors that are **uniquely** presented in one LRM's output.

## Chain-of-thought Format for Updating LOT when using PoR

-----
[Starts with summarizations of outputs, and a side-by-side comparison]
### [Existing reasoning behavior's name]
### [Definition of this reasoning behavior (reasoning behavior); reasoning with the given output; which output shows this reasoning behavior (with a quote)]
### [Whether this reasoning behavior occurs in OUTPUT A: Either "This reasoning behavior is observed in OUTPUT A." or "This reasoning behavior is not observed in OUTPUT A." DO NOT USE ANY OTHER EXPRESSIONS OR ADD OTHER DETAILS.]
### [Whether this reasoning behavior occurs in OUTPUT B: Either "This reasoning behavior is observed in OUTPUT B." or "This reasoning behavior is not observed in OUTPUT B." DO NOT USE ANY OTHER EXPRESSIONS OR ADD OTHER DETAILS.]

...{Repeat for the rest of existing reasoning behaviors in the reasoning taxonomy}...

{If you observe the distinguishing reasoning behaviors that are not in the reasoning taxonomy}
{Add new distinguishing reasoning behaviors}
### [New distinguishing reasoning behavior's name]
### [Definition of this reasoning behavior (reasoning behavior); reasoning with the given output; which output shows this reasoning behavior (with a quote)]
### [Whether this new reasoning behavior occurs in OUTPUT A: Either "This reasoning behavior is observed in OUTPUT A." or "This reasoning behavior is not observed in OUTPUT A." DO NOT USE ANY OTHER EXPRESSIONS OR ADD OTHER DETAILS.]
### [Whether this new reasoning behavior occurs in OUTPUT B: Either "This reasoning behavior is observed in OUTPUT B." or "This reasoning behavior is not observed in OUTPUT B." DO NOT USE ANY OTHER EXPRESSIONS OR ADD OTHER DETAILS.]
### [Is this reasoning behavior and its definition really different from the existing reasoning behavior above? If not, then you shouldn't include this reasoning behavior in the Added Rule section of your final output.]
...{Repeat for the other new reasoning behaviors}...

Final output:
Added:
[Added distinguishing reasoning behavior name]: [Detailed reasoning behavior definition (reasoning behavior)] [Example of this behavior quoted from the given outputs; Use the model's actual name if you want to make a reference but do not use OUTPUT A or OUTPUT B]
------

Figure 20: The example chain-of-thought format that we provided to the LLM annotator when instructing it to generate the updates to the taxonomy.

## G  HUMAN-DEFINED REASONING TAXONOMY BASELINE

We use the reasoning taxonomy defined by Gandhi et al. (2025) as another baseline for comparing LOT's classification accuracy. The reasoning features from this taxonomy, along with their definitions, are provided in Table 3. The set of reasoning behaviors—verification, backtracking, subgoal-setting, and backward chaining—is also used by the other behavioral studies of LRM (Bogdan et al., 2025; Jiang et al., 2025).

In our baseline experiment, we provide this taxonomy to Llama3.3 and instruct it to annotate the reasoning traces with the same prompt used for LOT (see Figure 15 and Figure 18).

Table 3: Human-defined reasoning taxonomy used in baseline comparison.

| Feature Name | Definition |
|---|---|
| Verification | The model systematically checks each step of its solution against established rules or data. This behavior ensures the solution's accuracy and consistency within the given framework. It involves confirming calculations, assumptions, and outcomes to maintain integrity in problem-solving. Example: The model faces a complex algebraic equation. It analyzes: "I will verify each transformation of the equation by checking algebraic identities." The model checks every step, ensuring no errors in logical transition or simplification have occurred. By cross-checking results with verified examples, the model establishes confidence in its solution. Upon detecting a mismatch, the model revisits previous steps to correct any potential errors. |
| Backtracking | The model revisits earlier stages in its problem-solving process to explore alternative pathways or correct mistakes. It traces back decision points to find where it diverged from a successful path, allowing for adjustments and retries. Example: The model works on a logic puzzle and concludes: "My current approach seems incorrect. I will backtrack to the last decision point and try an alternative solution path." The model reassesses the point where its assumptions might have derailed its strategy, opting to pursue a different branch to reach the correct solution. |
| Subgoal-Setting | The model breaks down complex problems into smaller, manageable subgoals. This behavior involves creating intermediate steps or milestones that guide the progression toward the ultimate solution, enhancing focus and organization. Example: The model tackles a multistep calculus problem. It states: "To solve this integral, I will first determine the derivatives involved as subgoals." By decomposing the problem into smaller parts, the model ensures each component is addressed thoroughly. Completing each subgoal incrementally builds the foundation leading to the primary objective. |
| Backward Chaining | The model starts with the desired outcome and works backward to identify necessary conditions that must be met. This deductive approach traces back from the goal to the known data points, ensuring the path taken is logically sound. Example: The model encounters a logic-based challenge. It declares: "I will set the target conclusion first, then determine what premises would logically entail this result." By analyzing the final objective, the model identifies required antecedents and systematically works backward, ensuring seamless causality in its reasoning process. |

## H   Prompts for VML and Few-Shot Prompting

> ### Prompt for VML Classification
>
> You are an expert in qualitative research and grounded theory, and you are good at distinguishing the reasoning behaviors of different language models using a reasoning taxonomy.
>
> You will be given two models' reasoning traces toward a question.
>
> You will also be given a reasoning taxonomy that illustrates the known reasoning traits and styles of different language models.
>
> You should think of this reasoning taxonomy as a classification rulebook that illustrates a set of decision rules that predict the author model of a reasoning output based on the reasoning behaviors observed in the reasoning output.
>
> Your task is to classify which reasoning trace belongs to which model based on these decision rules.
>
> Think step by step.
>
> For example, given a reasoning taxonomy with N distinguishing reasoning behaviors, your step-by-step chain of thoughts should look like this:
> ------
> {Compare two reasoning OUTPUTs side-by-side. Write a detailed summary of your analysis}
> ### [Name of the reasoning behavior that is applicable to the given outputs]
> ### [Definition of this reasoning behavior; how this reasoning behavior classifies the model; reasoning with the given output; which output shows this reasoning behavior (with a quote)]
> ### [Whether this reasoning behavior occurs in OUTPUT A: Either "This reasoning behavior is observed in OUTPUT A." or "This reasoning behavior is not observed in OUTPUT A." DO NOT USE ANY OTHER EXPRESSIONS OR ADD OTHER DETAILS.]
> ### [Whether this reasoning behavior occurs in OUTPUT B: Either "This reasoning behavior is observed in OUTPUT B." or "This reasoning behavior is not observed in OUTPUT B." DO NOT USE ANY OTHER EXPRESSIONS OR ADD OTHER DETAILS.]
> ### [Because of this reasoning behavior, which output is likely generated by which model]
> ...
> ### [Name of the reasoning behavior that is applicable to the given outputs]
> ### [Definition of this reasoning behavior; how this reasoning behavior classifies the model; reasoning with the given output; which output shows this reasoning behavior (with a quote)]
> ### [Whether this reasoning behavior occurs in OUTPUT A: Either "This reasoning behavior is observed in OUTPUT A." or "This reasoning behavior is not observed in OUTPUT A." DO NOT USE ANY OTHER EXPRESSIONS OR ADD OTHER DETAILS.]
> ### [Whether this reasoning behavior occurs in OUTPUT B: Either "This reasoning behavior is observed in OUTPUT B." or "This reasoning behavior is not observed in OUTPUT B." DO NOT USE ANY OTHER EXPRESSIONS OR ADD OTHER DETAILS.]
> ### [Because of this reasoning behavior, which output is likely generated by which model]
>
> [Summarize the analysis above in a paragraph]
>
> Now, I will classify the author model based on the applicable reasoning behavioral differences.
>
> Because of the reasoning behaviors [Reasoning Behavior Name x_1] ... [Reasoning Behavior Name x_n], the author model of OUTPUT A is [author model name]. Because of the reasoning behaviors [Reasoning Behavior Name y_1] ... [Reasoning Behavior Name y_n], the author model of OUTPUT B is [author model name].
> ------
>
> Think step by step. Your final classification should not bias the order of possible models appeared in the user's prompt.
>
> Make sure your output chain of thoughts follow this format exactly (including the hashtags ###).
>
> Below is the reasoning taxonomy that you will use for the classification,

Figure 21: Instruction for making classification using Verbalized Machine Learning (Xiao et al., 2025). This instruction is adapted from the PoR annotation instruction, and we highlight their key differences in red. To sum up, VML's instruction require the LLM to perform classification based on the patterns observed in the given reasonings and its decision rules.

## Prompt for Updating VML Decision Rules

You are an expert in qualitative research and grounded theory, and you are good at distinguishing the reasoning behaviors of different language models.

There is a reasoning taxonomy, which you can think of it as a rulebook that outlines a set of classification rules that can distinguish the reasonings generated by various large language models based on their reasoning behaviors. Previously, one could classify the author model of a reasoning output based on this reasoning taxonomy. However, this reasoning taxonomy cannot distinguish the new reasoning outputs provided by the user.

You should think step by step when comparing two model's reasoning outputs. It is okay if an existing reasoning behavioral difference does not appear in the provided output. However, if there are reasoning patterns that contradict the existing reasoning behaviors, you need to update those reasoning behaviors accordingly.

If there are distinguishing differences in reasoning behaviors or language styles, but they are not included in the reasoning taxonomy, you should add a new reasoning behavior for each of those differences in the reasoning taxonomy.

When adding the new reasoning behavior, you should provide a short name of the reasoning behavior with its detailed definition, such as [Reasoning Behavior name]: [Definition of this reasoning behavior] [if this reasoning behavior is observed, which model generated this reasoning output?] [Example of this behavior quoted from the given outputs]. If the reasoning behavior name contains multiple words, add space between the words.

[Example of this behavior] can be a direct quote. Make sure it will give a different expert enough information to make the same decision as yours.

Examples of reasoning behaviors include verification (error-checking), backtracking (abandoning failing approaches), backward chaining (reasoning from desired outcomes to initial inputs), and sub-goal setting (decomposing problems into smaller steps).

Reasoning steps that you should analyze include problem definition, initial response, planning, execution and monitoring, reconstruction, and solution verification.

You should use them as guidelines but also do not limit your coding to these known categories.

Each rule should describe one classification rule that classify one model's reasoning output from the other (for example, if this reasoning behavior is observed, then the author model is [model name]). Make sure you mention which model exhibits that reasoning behavior clearly.

For example, your step-by-step chain-of-thoughts should look like this: {chain-of-thought format}

Make sure you follow the exact format above when giving the added reasoning behavior. Write the reasoning behavior name, reasoning behavior definition, model exhibits that reasoning behavior, and example in the same line.

Below is the existing reasoning taxonomy,

Figure 22: Instruction for updating the decision rules of VML. This instruction is adapted from PoR's update instruction (differences highlighted in red), and a key difference is that the instruction asks the LLM to output if-else style decision rules for classifying an output's source LRM.

---

## Chain-of-thought Format for VML Update

------
[Starts with summarizations of outputs, and a side-by-side comparison]
### [Existing reasoning behavior's name]
### [Definition of this reasoning behavior; if this reasoning behavior is observed, which model generated this reasoning output; reasoning with the given output; which output shows this reasoning behavior (with a quote)]
### [Whether this reasoning behavior occurs in OUTPUT A: Either "This reasoning behavior is observed in OUTPUT A." or "This reasoning behavior is not observed in OUTPUT A." DO NOT USE ANY OTHER EXPRESSIONS OR ADD OTHER DETAILS.]
### [Whether this reasoning behavior occurs in OUTPUT B: Either "This reasoning behavior is observed in OUTPUT B." or "This reasoning behavior is not observed in OUTPUT B." DO NOT USE ANY OTHER EXPRESSIONS OR ADD OTHER DETAILS.]
### [Because of this reasoning behavior, which output is likely generated by which model]

...{Repeat for the rest of existing reasoning behaviors in the reasoning taxonomy}...

{If you observe the distinguishing reasoning behaviors that are not in the reasoning taxonomy}
{Add new reasoning behaviors for distinguishing reasoning traits and styles}
### [New reasoning behavior's name]
### [Definition of this reasoning behavior; if this reasoning behavior is observed, which model generated this reasoning output; reasoning with the given output; which output shows this reasoning behavior (with a quote)]
### [Whether this new reasoning behavior occurs in OUTPUT A: Either "This reasoning behavior is observed in OUTPUT A." or "This reasoning behavior is not observed in OUTPUT A." DO NOT USE ANY OTHER EXPRESSIONS OR ADD OTHER DETAILS.]
### [Whether this new reasoning behavior occurs in OUTPUT B: Either "This reasoning behavior is observed in OUTPUT B." or "This reasoning behavior is not observed in OUTPUT B." DO NOT USE ANY OTHER EXPRESSIONS OR ADD OTHER DETAILS.]
### [Because of this new reasoning behavior, which output is likely generated by which model]
### [Is this reasoning behavior and its definition really different from the existing reasoning behavior above? If not, then you shouldn't include this reasoning behavior in the Added Reasoning Behavior section of your final output.]
...{Repeat for the other new reasoning behaviors}...

Final output:
Updated:
[Original reasoning behavior name] -> [Updated reasoning behavior name]: [Updated detailed reasoning behavior definition] [If this reasoning behavior is observed, which model generated this reasoning output] [Updated example of this behavior quoted from the given outputs; Updated model's actual name if you want to make a reference]

Added:
[Added reasoning behavior name]: [Detailed reasoning behavior definition] [If this reasoning behavior is observed, which model generated this reasoning output] [Example of this behavior quoted from the given outputs; Use the model's actual name if you want to make a reference but do not use OUTPUT A or OUTPUT B]
------

Figure 23: The example chain-of-thought format that we provided to the LLM used in VML update. The format is adapted from the one used in PoR update (differences highlighted in red).

## Prompt for Few-Shot Classification

You are an expert in qualitative research and grounded theory, and you are good at distinguishing the reasoning behaviors of different language models using examples.

You will be given two models' reasoning traces toward a question, and examples of reasoning traces generated by these two models. Your task is to classify which reasoning trace belongs to which model, based on your observation of the few shot examples.

Think step by step. Compare the outputs against the examples provided in the system message.

Your final output should be:

------
{Compare the given reasoning OUTPUTs with all examples provided in this system message. See which model's outputs does OUTPUT A resembles the most. Then, check which model's outputs does OUTPUT B resembles the most.}

Based on my analysis above, the author model of OUTPUT A is [author model name], and the author model of OUTPUT B is [author model name].
------

Think step by step. Your final classification should not bias the order of possible models appearing in the user's prompt.

Make sure your output chain of thoughts follow this format exactly (including the hashtags ###).

Below are the few shot examples that you will use for the classification,

{Few-shot examples}

Figure 24: Instruction used in few-shot prompting baseline. Each shot of example contains a reasoning trace from each model that is in the comparison. The example reasoning traces are labeled with their source LRMs.

# I  PROMPTS FOR SUMMARIZING, MODIFYING, AND RE-EXPANDING REASONING STEPS

> **Prompt for Summarizing the Original Reasoning Trace**
>
> You will be provided with a snippet of a model's reasoning step towards a question. Your task is to summarize this reasoning step. Your summary should only include the key information of the reasoning step so one can reconstruct the reasoning step by filling in the details of your summary. Please only include the summary in your output. Do not add any other details.
>
> Below is the reasoning step that you need to summarize:
> {Paragraph of original reasoning}

Figure 25: Prompt used in having a Qwen3 model to summarize the paragraphs of its original reasoning traces into a high-level summary

> **Prompt for Modifying the Original Reasoning Trace**
>
> You will be given a summary of a reasoning trace. Your task is to modify this reasoning traces based on the reasoning guideline below.
>
> You need to add or subtract reasoning steps (numbered items) to the original summary based on the guidelines.
>
> Your guidelines are:
> To improve the model's chance of answering this question correctly, in their reasoning:
> {Reasoning behaviors with odds ratio of its reasoning traces being correct > 1}
>
> Finally, the model should avoid the following reasoning behaviors in their thinking:
> {Reasoning behaviors with odds ratio of its reasoning traces being correct < 1}
>
> Modify the reasoning summary below:
> {Reasoning summary}
>
> Output your modified summary after "Modified Summary:". Do not add any other details in your output.

Figure 26: Prompt used in having a Qwen3 model to modified a list of reasoning steps summarized from the paragraphs of its original reasoning traces.

> **Prompt for Expanding the Modified / Unmodified Summary**
>
> You will be given a summarized reasoning step toward a question. Your task is to fill in the details of that summarized reasoning. {If it is not the last reasoning step in the summary: However, you don't need to give me your final answer to the question.} You can use the question prompt as a reference when filling the details of this step.
>
> The question for your reference is: {Question prompt}
>
> For your context, below are the expanded previous reasoning steps, prior to the step that you need to fill in the details: {Expanded previous reasoning steps}
>
> Below is the summarized reasoning step:
> {Summarized reasoning step}
>
> Fill in the details of the summarized reasoning step above. You don't need to solve this question. Your task is to fill in the details of the summarized reasoning step. Your output should be an expanded reasoning step with details.

Figure 27: Prompt used in having a Qwen3 model to re-expand a summaried reasoning step into a full reasoning paragraph given its previous expanded reasoning steps and question prompt as context.

| Qwen3-0.6B Odds Ratio | |
|---|---|
| Verifying Method's Applicabaility | 2.26 |
| Evaluating Problem's Complexity | 1.51 |
| Comprehensively Applying Principles | 1.40 |
| Flexible Approach Adaptation | 1.25 |
| Oversimplicifying the Problem | 1.15 |
| Considering Problem Context | 1.11 |
| Inconsistent Application of Principles | 0.76 |
| Redundant Evaluation of the Same Infor | 0.71 |

| Qwen3-1.7B Odds Ratio | |
|---|---|
| Verifying Solution Against Given Options | 2.78 |
| Choosing Mathematical Tools Based on Context | 2.33 |
| Step-by-Step Derivation | 2.22 |
| Applying Relevant Theories | 1.67 |
| Identification of Key Information | 1.50 |
| Elinimating Impossible Options | 1.39 |
| Exploring Multiple Approaches | 1.05 |
| Generating Hypothesis from Existing Knowledge | 0.00 |

| Qwen3-4B Odds Ratio | |
|---|---|
| Direct Application of Formulas | 2.58 |
| Formula Manipulation And Analysis | 2.58 |
| Evaluating Problem Constraints | 1.77 |
| Retrieving Relevant Physical Principles | 1.52 |
| Scrutinizing Assumptions | 1.26 |
| Adjusting Hypothesis Based on New Information | 1.22 |
| Cyclical Reasoning | 1.11 |
| Exhaustive Hypothesis Generation | 0.98 |

| Qwen3-8B Odds Ratio | |
|---|---|
| Systematically Resolving Uncertainties | 1.83 |
| Error Recognition and Recovery | 1.13 |
| Constantly Re-evaluating Problem Solving Approach | 1.13 |
| Overly Broad Conceptualization | 0.92 |
| Jumping between Concepts | 0.81 |
| Applying Chemical Knowledge | 0.75 |
| Inconsistent Application of Chemical Principles | 0.54 |
| Visualize Molecular Structure of Compounds | 0.32 |

Figure 28: Odds ratios for all reasoning differences observed between Qwen3-0.6B/1.7B/4B/8B and Qwen3-32B using PoR representations.

## J    ASSOCIATION BETWEEN REASONING DIFFERENCE AND MODEL PERFORMANCES

We report the odds ratios $\frac{p(x \in \text{correct}|x_c=1)/p(x \in \text{wrong}|x_c=1)}{p(x \in \text{correct}|x_c=0)/p(x \in \text{wrong}|x_c=0)}$ for all reasoning differences observed between Qwen3-0.6B/1.7B/4B/8B and Qwen3-32B in Figure 28 on GPQA dataset.

For most reasoning differences, if it is more frequently observed in Qwen3-32B's reasoning, its occurrence in the smaller Qwen models tends to be more strongly associated with correct reasoning (odds ratio $> 1$). For example, *"verifying solutions against given options"* appears about three times more often in Qwen3-32B's reasoning traces than in those of Qwen3-1.7B, and its odds ratio for Qwen3-1.7B is 2.78, meaning the odds of a correct answer are 2.78 times higher when this feature is present. In contrast, reasoning traits of the smaller models more often have odds ratios smaller than or close to 1, suggesting they contribute little to correctness and in some cases are more associated with incorrect reasoning.

There are only three exceptions. First, *"generating hypotheses from existing knowledge"* has a zero odds ratio for Qwen3-1.7B, partially because Qwen3-1.7B only exhibits this trait once in its reasoning. Nonetheless, this behavior indeed has a $> 1$ (1.2) odds ratio on Qwen3-32B's outputs.

The other two exceptions are observed on Qwen3-8B: the *"applying chemical knowledge"* ($c_{\text{apply}}$) has an odds ratio of 0.75, mostly because this behavior often co-occurs with *"inconsistent application of chemical principles"* ($c_{\text{inconsistent}}$ and $p(c_{\text{inconsistent}}|c_{\text{apply}}) = 0.45$), weakening its association with correctness.

Similarly, visualizing molecular structures is more strongly associated with incorrect reasoning, despite being more common in Qwen3-32B's reasoning. However, this behavior also shows a lower than 1 (0.43) odds ratio for Qwen3-32B. This suggests that, although visualizing compound structures reflects an advanced reasoning behavior, it does not reliably contribute to correctness. Indeed, given the limited expressiveness of text, accurately representing complex chemical structures (e.g., rings) in plain text is challenging.

## K    USE OF LARGE LANGUAGE MODELS

We used large language models only to polish the grammar of our writing. They were not used for research ideation or for retrieving related works.

