# OpenReview forum: "Your thoughts tell who you are: Characterize the reasoning patterns of LRMs"
_ICLR.cc/2026/Conference — ICLR 2026 Conference Withdrawn Submission_

### Official Review · Reviewer_YxvB · 2025-10-29

**Soundness:** 3
**Presentation:** 3
**Contribution:** 3
**Rating:** 4
**Confidence:** 4

**Summary:**

This paper introduces the LLM-proposed Open Taxonomy (LOT), an inductive method for systematically characterizing and distinguishing the reasoning patterns of Large Reasoning Models (LRMs). LOT uses an LLM to iteratively generate a human-readable taxonomy of reasoning features by comparing reasoning traces from two models. The method is extensively evaluated on 12 open-source LRMs across math, science, and coding tasks, achieving 80-100% accuracy in distinguishing models based on scale, family, or domain. Furthermore, the authors demonstrate a causal link between reasoning style and performance by modifying the reasoning patterns of smaller models, leading to a 3.3-5.7% accuracy improvement on the GPQA benchmark.

**Strengths:**

1. The LOT framework is intestering. Its inductive, data-driven approach  overcomes the limitations of predefined, deductive taxonomies, allowing it to uncover unexpected reasoning behaviors.
2. The experimental design is comprehensive. The paper evaluates a wide range of 12 LRMs, varying in scale, base model family, and task specialization, across five diverse benchmarks. This breadth strongly supports the generalizability of the findings.

**Weaknesses:**

1. While the paper shows LOT can distinguish models and that reasoning affects performance, it does not establish a direct causal link between model meta-attributes (e.g., parameter count, training data) and their resulting reasoning patterns. Future work could involve controlled experiments (e.g., ablations on training data) to strengthen this causal inference.
2. Although Appendix D addresses stability across random seeds, the probabilistic nature of the LLM-based feature generation remains a potential source of variation. The main text could more explicitly discuss the convergence properties and the potential impact of the annotator LLM's randomness on the final taxonomy.
3. The current intervention method (summarize → edit → re-expand) is somewhat complex and computationally intensive. It also fails for Qwen3-1.7B. Exploring more lightweight and direct methods for steering reasoning at test-time would be a valuable direction for future research.
4. All experiments rely solely on Llama3.3-70B-Instruct as the annotator. The impact of this choice on the generated taxonomies is not explored. A brief analysis or discussion on how LOT's performance might vary with the choice of the annotator LLM would strengthen the methodology.

**Questions:**

1.The intervention study compellingly shows that modifying reasoning can improve performance. However, to more firmly establish that the meta-attributes (like model scale) directly cause specific reasoning patterns, have you considered or are there plans for controlled experiments?
2.For instance, comparing models trained with identical data but different scales, or ablating specific training data components to isolate their effect on the emergent reasoning style?
3. The summarization-based intervention pipeline significantly hurt the performance of Qwen3-1.7B, leading to overall worse results despite the modification. What is your hypothesis for this specific failure?  Does it relate primarily to its poorer instruction-following capability, as noted in Appendix C, or is the summarization-and-expansion process itself too demanding for such a small model?

---

### Official Review · Reviewer_5YzK · 2025-11-01

**Soundness:** 2
**Presentation:** 2
**Contribution:** 2
**Rating:** 4
**Confidence:** 4

**Summary:**

This paper introduces LLM-proposed Open Taxonomy (LOT), an inductive framework that uses LLMs to generate, annotate, and refine human-readable reasoning features distinguishing different large reasoning models (LRMs). LOT compares reasoning traces pairwise, iteratively expands its feature set, and uses a logistic classifier to predict the source model.

**Strengths:**

The paper takes a creative approach to understanding how large reasoning models think. I like that the method is interpretable and that the taxonomy the authors propose feels intuitive rather than overly technical. The experiments are thorough and carefully designed, and the results are presented in a clear way. Overall, it's a timely and relevant paper that provides useful insights into the diversity and structure of reasoning patterns in modern LLMs.

**Weaknesses:**

baseline comparison: The baseline comparison isn’t fully fair. LOT is trained with a classifier, but the other methods like few-shot prompting and VML are not, which naturally gives LOT an advantage. It would be better to also compare with trained but non-interpretable methods, like a simple classifier on embeddings (similar to [1]).  Also, the few-shot baseline could be improved by letting the model reason first and then decide.


[1] Mingjie Sun, Yida Yin, Zhiqiu Xu, J Zico Kolter, and Zhuang Liu. Idiosyncrasies in large language models. International Conference on Machine Learning, 2025.


Generalization of features:  The paper doesn’t test whether LOT’s discovered features generalize across datasets or reasoning domains. The features are extracted and evaluated within the same dataset they were generated from, so it’s unclear if they reflect true reasoning behaviors or just surface-level patterns tied to specific tasks. Testing whether the same features remain predictive across different datasets would strengthen the claim that LOT captures general reasoning traits rather than dataset artifacts.


Causal argument: Causal claim is not well supported. The intervention only modifies reasoning text at test time, which changes many factors beyond the targeted behaviors, making it unclear whether the improvements truly come from the reasoning features or from unrelated text effects. The analysis remains correlational rather than causal

**Questions:**

Line 266: Why did you cite Bai et al., 2022 for few-shot prompting?

---

### Official Review · Reviewer_R9Ky · 2025-11-01

**Soundness:** 2
**Presentation:** 3
**Contribution:** 2
**Rating:** 2
**Confidence:** 4

**Summary:**

This paper introduces the LLM-proposed Open Taxonomy (LOT), a method to compare reasoning traces coming from different language models. The method operates in three stages: first, an LLM is used to compare reasoning traces from two models on the same input and provides their characteristic differences. Second, the characteristics are identified in reasoning traces for different inputs, and third, a classifier is trained to predict the source model for unseen reasoning traces.

The paper applies LOT to 12 open-source reasoning LLMs on five datasets: GPQA-Diamond, MATH-500, AIME-24/25, CRUXEVAL and LiveCodeBench. Llama-3.3-70B-Instruct is used as the annotation model and LOT is compared to few-shot prompting baselines. Experimental results show that LOT can accurately predict the sources of reasoning traces. The authors also associate reasoning differences to differences in task performance and use that insight to boost task performance for certain models.

**Strengths:**

This paper introduces a method to classify reasoning traces w.r.t. their source model. It provides extensive experiments covering 12 models and five datasets. The experimental insights attributing reasoning differences to differences in task performance are interesting and can lead to a better understanding of how reasoning behaviour can be further optimised in LLMs.

**Weaknesses:**

The paper’s technical contributions and novelty are limited. The proposed approach uses a single LLM to generate features from reasoning traces which are then used to train a logistic classifier. While this is an interesting approach it currently is somewhat narrowly scoped and would require additional analysis for acceptance to the main conference. For example:
* Experiments limited to a single model: while acknowledged by the authors, the paper uses a single model for feature generation (Llama-3.3-70B-Instruct) and the authors do not discuss the impact of using different models to power LOT.
* Lack of LLM verification: the features generated by the LLM in LOT have not been verified via human quality checks. To establish LOT as a reliable method to generate characteristics of reasoning traces it would be necessary to provide human assessments and verification of the method’s correctness.

**Questions:**

None

---

### Note · Authors · 2025-11-18

I have read and agree with the venue's withdrawal policy on behalf of myself and my co-authors.